# Incentives in Federated Learning: Equilibria, Dynamics, and Mechanisms for Welfare Maximization

Aniket Murhekar[1]    Zhuowen Yuan[1]    Bhaskar Ray Chaudhury[1]    Bo Li[1]    Ruta Mehta[1]

[1]University of Illinois, Urbana-Champaign

{aniket2,zhuowen3,bryacha,lbo,rutameht}@illinois.edu

## Abstract

Federated learning (FL) has emerged as a powerful scheme to facilitate the collaborative learning of models amongst a set of agents holding their own private data. Although the agents benefit from the global model trained on shared data, by participating in federated learning, they may also incur costs (related to privacy and communication) due to data sharing. In this paper, we model a collaborative FL framework, where every agent attempts to achieve an optimal trade-off between her learning payoff and data sharing cost. We show the existence of Nash equilibrium (NE) under mild assumptions on agents' payoff and costs. Furthermore, we show that agents can discover the NE via best response dynamics. However, some of the NE may be bad in terms of overall welfare for the agents, implying little incentive for some fraction of the agents to participate in the learning. To remedy this, we design a *budget-balanced mechanism* involving *payments* to the agents, that ensures that any $p$-mean welfare function of the agents' utilities is maximized at NE. In addition, we introduce a FL protocol FedBR-BG that incorporates our budget-balanced mechanism, utilizing best response dynamics. Our empirical validation on MNIST and CIFAR-10 substantiates our theoretical analysis. We show that FedBR-BG outperforms the basic best-response-based protocol without additional incentivization, the standard federated learning protocol FedAvg (McMahan et al. [2017]), as well as a recent baseline MWFed (Blum et al. [2021]) in terms of achieving superior $p$-mean welfare.

## 1   Introduction

Federated Learning (FL) has emerged as an effective collaborative training paradigm, where a group of agents can jointly train a common machine learning model. The success of collaborative learning paradigms is visible in the domains of autonomous vehicles (Elbir et al. [2020]), digital healthcare (Dayan et al. [2021], Xu et al. [2021], Nvidia [2019]), multi-devices (Learning [2017]), and biology (Bergen and Petryshen [2012]). Although such collaborative learning is immensely beneficial to the agents, individually, they may not be incentivized to share their data. This is because sharing data may incur costs attributed to bandwidth use, privacy leakage, and the use of computing resources. In turn, high costs and low learning payoffs may cause some of the agents to drop out of FL, resulting in subpar learning. As noted by Blum et al. [2021], the success of FL systems depends on its ability to attract and retain a large number of federating agents.

   *Thus, it is crucial to achieve fairness and welfare guarantees for all participating agents.*

This calls for game-theoretic modeling and analysis of the agent's payoff and costs, and subsequent mechanism design to incentivize participation. Towards the former, Blum et al. [2021] introduce a model, where every agent receives a *payoff* from the collaboration measuring the "learning benefit" the agent derives from the total data shared. In particular, when each agent $i$ contributes $s_i$ units of data, the payoff for agent $i$ is captured by $a_i(\boldsymbol{s})$, where $\boldsymbol{s} = \langle s_1, s_2, \ldots, s_n \rangle$ is the data contribution

37th Conference on Neural Information Processing Systems (NeurIPS 2023).

profile, or *sample vector*. Blum et al. [2021] consider the *constrained cost-minimizing model*, where each agent minimizes her data contribution ($s_i$) subject to the requirement that her payoff should be larger than a threshold ($a_i(s_i, \boldsymbol{s}_{-i}) \geq \mu_i$). They show that a Nash equilibrium (NE) (Nash [1951]), arguably the most sought after solution concept in game theory, may not always exist in this model. They derive sufficient conditions for existence of NE, as well as provide novel structural results about the equilibria.

We note that an agent's cost for sharing data may be more complex than just the size of the total data shared. There are studies dedicated to quantifying the losses (attributed to data collection, processing, and privacy) that are incurred with increased sharing of data (Li and Raghunathan [2014], Laudon [1996], Jaisingh et al. [2008]). Furthermore, within game theory (and economics) typically agents are considered as net utility maximizers, *i.e.,* maximizing payoff minus cost, for example value minus payment in auctions (Myerson [1981]), and revenue minus cost in markets (Huber et al. [2001], Börgers [2015]). To capture these aspects, in this paper we propose an *unconstrained utilitarian model*, where we define the *utility* $u_i(\boldsymbol{s})$ of each agent $i$ as the difference between her payoff and her cost of sharing data, i.e., $u_i(\boldsymbol{s}) = a_i(\boldsymbol{s}) - c_i(s_i)$, where $c_i(s_i)$ denotes the cost incurred by agent $i$ on sharing $s_i$ data samples. Each agent aims to maximize their utility. This model is natural in FL settings where there is a natural coupling between payoffs and costs, and there are no hard cost constraints, i.e., having an upper bound on the total cost that an agent can incur. Since utility functions are directly indicative of the welfare for the agents, this facilitates defining global fairness and welfare criteria in terms of the utilities of the agents. This motivates the following questions:

When agents strategize on data contribution to maximize their net utility, does a Nash equilibrium always exist? If yes, then can agents actually *discover a Nash equilibrium* when acting independently? Can any welfare function of utilities of the agents be optimized at such a Nash equilibrium by *designing new rules* (*mechanisms*)?

The goal of this paper is to address the above questions. Before we describe our contributions, we note that Karimireddy et al. [2022] also consider a framework similar to ours, by defining the utility function to be payoff minus cost. However, their focus is *data maximization* and avoiding *free-riding*, while the focus of our paper is to design mechanisms that achieve fairness and welfare for all agents. Furthermore, our model generalizes the model of Karimireddy et al. [2022] as we do not require the agents' data to be identically distributed and in turn to have identical payoff functions.

## 1.1 Our Contributions

We work under the *concavity assumption* on the utility functions. In particular, we assume that the payoff functions are concave and cost functions are convex. These assumptions are standard in the literature (Blum et al. [2021], Karimireddy et al. [2022]). Convexity of cost functions is a natural choice since it captures the property of increasing marginal costs. For instance, data sharing through *ordered selection*, i.e., sharing records in ascending order of costs involved for collecting the records, results in convex cost functions. There are more models that result in strictly convex cost functions (Li and Raghunathan [2014]). Similarly, several important ML models exhibit concave payoff functions; for instance payoffs in linear or random discovery models, random coverage models, and general PAC learning are all concave – see Section 2.1. Furthermore, there is empirical evidence that the accuracy function in neural networks under the cross-entropy loss is also concave (Kaplan et al. [2020]).

**Existence and Reachability of Nash Equilibrium:** Under the concavity assumptions, we show that a Nash equilibrium (NE) is guaranteed to exist. We note that our existence result holds for more general settings than those of Blum et al. [2021] – we do not assume that the cost functions are linear, or that the utility functions have bounded derivatives. In particular, our result shows the existence of NE under the general PAC learning framework (see Section 2.1); under this setting an equilibrium in the model of Blum et al. [2021] fails to exist. We also demonstrate that if the concavity assumption is relaxed, there exist instances which do not admit NE.

Furthermore, we show that the agents can *discover* a NE through an intuitive *best response dynamics*, where agents update their data contribution proportional to the gradient of their utility functions in the direction that increases their utility. We show that for strongly concave utility functions and under the mild assumption of the gradient of the utilities being bounded, the best response dynamics converges

to an approximate NE in time polynomial in $O(\log(\varepsilon^{-1}))$. This contribution may be of independent interest to equilibrium computation in concave games.

**A Fair and Welfare Maximizing Mechanism:** In general, a Nash equilibrium need not be fair or maximize any notion of welfare for the agents (see Example 1). Therefore, we next focus on designing mechanisms that optimize welfare of the agents at its NE. The generalized $p$-mean welfare, defined to be the $p^{th}$-norm of the utilities of the agents $(\frac{1}{n}\sum_{i\in[n]} u_i(s)^p)^{\frac{1}{p}}$ for $p \in (-\infty, 1]$, constitutes a family of functions characterized by natural fairness axioms including the Pigou-Dalton principle (Moulin [2003]). This notion encompasses well studied notions of fairness and economic welfare, such as (i) the average social welfare $(\frac{1}{n}\sum_{i\in[n]} u_i(s))$ when $p = 1$, (ii) the egalitarian welfare $(\min_{i\in[n]} u_i(s))$ when $p = -\infty$, which is a fundamental measure of fairness, and (iii) the Nash welfare $(\prod_{i\in[n]} u_i(s))^{1/n}$ in the limiting $p \to 0$ case, which is a popular notion in social choice theory that achieves a balance between welfare and fairness (Varian [1974], Caragiannis et al. [2019]).

As our second main contribution, for linear costs, we design a *budget-balanced mechanism* that always admits a Nash equilibrium. The mechanism involves *payments*, however, budget-balancedness ensures that the total payment of all the agents is zero, and thereby the central server operates on no-profit-no-loss. Moreover we show that when the sample vector at NE is positive, i.e., all agents contribute positive amount of data samples, then the NE maximizes the $p$-mean welfare among all positive sample vectors. Since we can ensure the server does not communicate with an agent who does not contribute any data points, insisting on positive sample vectors is a mild assumption.

Our mechanism builds on ideas from a mechanism of Falkinger et al. [2000] for the efficient provisioning of public goods. The key idea is to compensate an agent who incurs a cost higher than the average cost incurred by other agents via a subsidy proportional to her excess cost; likewise, agents incurring a lower cost than average of others are proportionally taxed. By setting the level of tax/subsidy carefully, we show that the NE of the mechanism are $p$-mean welfare maximizing. To the best of our knowledge, our work is the first to explore this intimate connection between FL and public goods provisioning. Our results highlight the promise of bridging the fields of machine learning and social choice theory.

Once we have established the mechanism, we show that a corollary of our first main result implies that the *best response dynamics* under this mechanism will lead to the discovery of the $p$-mean welfare maximizing Nash equilibrium.

**Empirical Evaluation:** We design a distributed training protocol for FL based on our mechanism, called FedBR-BG. We compare our algorithm with three other algorithms: the distributed protocol for the vanilla mechanism without budget balancing FedBR, the standard federated learning protocol FedAvg (McMahan et al. [2017]), and a recent adaptation of FedAvg called MWFed (Blum et al. [2021]). We show that our budget-balancing algorithm achieves superior $p$-mean welfare under different settings on two datasets, MNIST (LeCun et al. [2010]) and CIFAR-10 (Krizhevsky [2009]).

## 1.2   Related Work

We mention some additional literature on welfare maximization and incentives in FL, as well as other related mechanisms in public goods theory.

**Welfare maximization in FL.** Typically federated learning protocols compute a model which maximize some weighted sum of agent accuracies (i.e. utilties). Examples of such protocols include the standard FedAvg (McMahan et al. [2017]), AFL (agnostic FL) (Mohri et al. [2019]), and $q$-FFL (Li et al. [2020]). However, unlike our work, all these methods ignore the strategic aspects arising from costs involved in data sharing, and instead assume agents honestly contribute all their data.

**Incentives in FL.** This line of work adopts game theory for incentivizing the contribution of data owners. Common models include the Stackelberg game, non-cooperative game, and coalition game. More specifically, the Stackelberg game is employed to optimize the utility of both the server and agents Feng et al. [2019]. On the other hand, in non-cooperative games, the server or the agent seeks to maximize its own utility Zou et al. [2019]. While most previous methods analyze the properties of a certain scenario, we aim to design mechanisms that achieve fairness and welfare for all agents.

**Mechanisms for public good provisioning.** There is a long line of work for the efficient provisioning of public goods, beginning from Samuelson [1954]. Several works such as Falkinger et al. [2000], Andreoni and Bergstrom [1996] and Bergstrom et al. [1986] use the idea of imposing a tax/subsidy on the agents but differ in the specific manner in which this tax/subsidy is imposed. While our mechanism is inspired by Falkinger et al. [2000], our model is more general than theirs. The design of our mechanism for this general model and the proof of its properties are our novel contributions.

## 2 Problem formulation

We consider a federated learning problem with $n$ agents who wish to jointly solve a common learning problem. Let $\mathcal{D}_i$ be the distribution of data points available to agent $i$. Towards jointly solving the learning problem, each agent $i$ contributes some set $T_i \sim \mathcal{D}_i^{s_i}$ of $s_i$ data samples. Under the assumption that agent $i$'s data is i.i.d. from their distribution $\mathcal{D}_i$, each agent's contribution can be captured simply by their contribution level, i.e., amount of data samples they contribute. Let $S_i \subseteq \mathbb{R}_{\geq 0}$ be the set of feasible contribution levels of agent $i$, and let $\mathcal{S} := \bigtimes_j S_j$. Given a *sample vector* $\boldsymbol{s} = (s_1, \ldots, s_n) \in \mathcal{S}$, the central server returns model(s) trained using the samples $\bigcup_i T_i$.

In our model, each agent $i$ derives a *payoff* from the jointly learned model, e.g. the payoff could be the accuracy of the model. We assume a general framework which models the payoff of agent $i$ as a function $a_i : \mathcal{S} \to \mathbb{R}_{\geq 0}$. We typically assume each payoff function $a_i$ is bounded, continuous in $\boldsymbol{s}$, and non-decreasing in the contribution $s_i$ of agent $i$. Moreover each agent $i$ incurs a *cost* associated with data sharing captured through a non-decreasing cost function $c_i : S_i \to \mathbb{R}_{\geq 0}$. The net utility of agent $i$ is the payoff minus cost, *i.e.,*

$$u_i(\boldsymbol{s}) = a_i(\boldsymbol{s}) - c_i(s_i).$$

Given the above framework, the goal of an agent $i$ is to decide how many samples to contribute so that her net utility $u_i(\cdot)$ is maximized. Note that the utility of agent $i$ depends on the contributions of other agents as well. Further, we can assume without loss of generality that each set $S_i = [0, \tau_i]$ for some threshold $\tau_i > 0$. This is because we can discard contribution levels where an agent gets negative utility. Since the payoff $a_i(\boldsymbol{s})$ is bounded above by some constant $A_i > 0$ and costs are increasing, agent $i$ cannot obtain a positive utility by contributing more than $\tau_i := c_i^{-1}(A_i)$ samples.

Payoff and cost functions are assumed to be concave and convex respectively (Blum et al. [2021], Karimireddy et al. [2022]). As discussed in Section 1.1, it is natural to assume that cost functions are convex to capture increasing marginal costs (Li and Raghunathan [2014]). Similarly, there are ample justifications to concave payoff functions, as discussed in Section 2.1 where we analyze payoff functions arising from some of the canonical learning paradigms, and their *concavity properties*.

*Remark* 1. Our framework generalizes those of Blum et al. [2021] and Karimireddy et al. [2022]. In the former, an agent's goal is to contribute the fewest number of data samples subject to ensuring that their payoff crosses a certain threshold; and in the latter all agents have a common payoff function that is a function of $\|s\|_1 = s_1 + \cdots + s_n$, and linear cost functions.

**Nash Equilibrium (NE).** Arguably the most sought after solution concept within game theory is of *Nash Equilibrium* (Nash [1951]), a stable state or an equilibrium state of the system where no agent gains by unilaterally changing their data contribution level. Formally,

**Definition 1** (Nash equilibrium (NE))**.** A sample vector $\boldsymbol{s} \in \mathcal{S}$ is said to be at a Nash equilibrium if for every $i \in [n]$, and every $s_i'$, we have $u_i(\boldsymbol{s}) \geq u_i(s_i', \boldsymbol{s}_{-i})$ where $(s_i', \boldsymbol{s}_{-i}) = (s_1, \ldots, s_i', \ldots, s_n)$.

An alternate view of a Nash equilibrium relies on the concept of *best response*. Given the sample contributions $\boldsymbol{s}_{-i}$ of other agents, the set $f_i(\boldsymbol{s}_{-i})$ of contribution levels of agent $i$ that maximize her utility is the best response set of agent $i$:

$$f_i(\boldsymbol{s}_{-i}) = \arg\max_{x \in S_i} \big\{ a_i(x, \boldsymbol{s}_{-i}) - c_i(x) \big\} \subseteq S_i.$$

The *best response correspondence* $f$ is then defined as a set-valued function $f : \mathcal{S} \to \bigtimes_i 2^{S_i}$, where $[f(\boldsymbol{s})]_i = f_i(\boldsymbol{s}_{-i})$. It is then clear that:

**Proposition 1.** *A sample vector $\boldsymbol{s} \in \mathcal{S}$ is a Nash equilibrium if and only if it is a fixed point of the best response correspondence, i.e., $\boldsymbol{s} \in f(\boldsymbol{s})$.*

## 2.1 Canonical examples of payoff functions

We now discuss a few examples of payoff functions that are captured by our general framework. In all the examples below, the payoff functions $a_i(\boldsymbol{s})$ are non-negative, bounded, continuous, non-decreasing, and concave in $s_i$ for any fixed strategy profile $\boldsymbol{s}_{-i}$ of the other agents.

**Linear or Random discovery.** In this model, the payoff is linear in the sample vector and is given by $a_i(\boldsymbol{s}) = (W\boldsymbol{s})_i$ for a matrix $W$. For example, Blum et al. [2021] consider a setting where each agent has a sampling probability distribution $\mathbf{q}_i$ over the instance space $X$ and gets a reward equalling $q_{ix}$ whenever the instance $x$ is sampled by any agent. Then the expected payoff to agent $i$ is $a_i(\boldsymbol{s}) = (QQ^T\boldsymbol{s})_i$, where $Q$ is a matrix given by $Q[i, x] = q_{ix}$ for $i \in [n]$ and $x \in X$. Here $W = QQ^T$ is a symmetric PSD matrix with an all one diagonal.

**Random coverage.** In the above setting, agent $i$ obtains a reward $q_{ix}$ each time some agent samples instance $x$. In the random coverage model arising in binary classification (Blum et al. [2021]), an agent gets this reward only once. Under this model, the payoff given by expected accuracy takes the form:

$$a_i(\boldsymbol{s}) = 1 - \frac{1}{2} \sum_{x \in X} q_{ix} \prod_{j=1}^{n} (1 - q_{jx})^{s_j} \in [0, 1]. \tag{1}$$

**Generalization bounds from general PAC learning.** Consider a general learning setting where we want to learn a model $h$ from a hypothesis class $\mathcal{H}$ which minimizes the error over a data distribution $\mathcal{D}$ given by $R(h) = \mathbb{E}_{(x,y) \in \mathcal{D}} e(h(x), y)$, for some loss function $e(\cdot)$. Given $m$ i.i.d. data points, the empirical risk minimizer (ERM) can be computed as the model $h_m = \arg\min_{h \in \mathcal{H}} \sum_{\ell \in [m]} e(h(x_\ell), y_\ell)$. Mohri et al. [2018] show the following bound on the error of $h_m$, which holds with high probability:

$$1 - R(h_m) \geq a(m) := a_0 - \frac{4 + \sqrt{2k(2 + \log(m/k))}}{\sqrt{m}}, \tag{2}$$

where $(1 - a_0)$ is the accuracy of the optimal model from $\mathcal{H}$, and $k$ is a (constant) measure of the difficulty of the learning problem depending on $e(\cdot)$ and $\mathcal{H}$. Using this we can define the agent payoff $a_i$ function as the accuracy of the learning task as $a_i(\boldsymbol{s}) = a(\|\boldsymbol{s}\|_1)$[1].

**Empirical evidence.** Kaplan et al. [2020] discuss empirical scaling laws relating the cross-entropy loss on neural language models. They observe that the loss scales with the dataset size $m$ as a power law $\ell(m) = \alpha \cdot m^{-\beta}$, for some parameters $\alpha > 0$ and $\beta \in (0, 1]$. This naturally defines the payoff function as the accuracy of the learning task as

$$a_i(\boldsymbol{s}) = 1 - \alpha_i \cdot \|\boldsymbol{s}\|_1^{-\beta_i}. \tag{3}$$

In addition, the pay-off functions of current large language models (e.g., accuracies) are also non-negative and non-decreasing as a function of the data size (Henighan et al. [2020]).

## 3 Nash Equilibrium: Existence and Best Response Dynamics

In this section, we investigate the existence and computation of a Nash equilibrium. We start with two positive results showing the existence of a Nash equilibrium for a broad class of payoff and cost functions.

**Theorem 3.1.** *A Nash equilibrium exists in any federated learning problem where for every agent $i$ the payoff function $a_i(\boldsymbol{s})$ is continuous in $\boldsymbol{s}$ and concave in $s_i$, and cost function $c_i$ is increasing and convex in $s_i$.*

*Proof.* We will show the existence of a Nash equilibrium by showing that the best response correspondence $f$ has a fixed point. First observe that $f$ is defined over a compact, convex domain $\mathcal{S} = \bigtimes_j S_j$ since each $S_j$ is convex. Next, note that agent $i$'s utility function $u_i(\boldsymbol{s}) = a_i(\boldsymbol{s}) - c_i(s_i)$ is continuous in $\boldsymbol{s}$ due to the continuity of $a_i$ and $c_i$. The continuity of $u_i$ in $\boldsymbol{s}$ and the concavity of

---

[1]We define $a(\mathbf{0}) = 0$.

$u_i$ in $s_i$ implies the upper semi-continuity of the best response correspondence $f_i$. Moreover, $u_i$ is concave in $s_i$ for each fixed $\boldsymbol{s}_{-i}$, since $a_i$ and $-c_i$ are concave in $s_i$. Thus for each fixed $s_{-i}$, the best response set $f_i(s_{-i}) \subseteq \mathbb{R}_{\geq 0}$ is a non-empty interval, and hence is also convex. Thus $f$ is a upper semi-continuous non-empty and convex valued correspondence defined over a compact, convex domain. By the Kakutani fixed-point theorem (Kakutani [1941]), $f$ admits a fixed point.

The above result can also be proved directly by invoking Rosen [1965], who showed the existence of a Nash equilibrium of $n$-person concave games, where the utility function of an agent $i$ is defined over closed, compact set, is continuous and concave in $i$'s own strategy. $\qquad\square$

**Implications.** Theorem 3.1 shows that a Nash equilibrium exists when payoffs are concave and costs are convex. All our motivating examples from Section 2.1 lie under this concave/convex regime and therefore admit a NE. In Appendix A we discuss existence and non-existence of Nash equilibrium in our model when we go beyond the concave-convex regime of payoff and cost functions. In particular, we show that Nash equilibrium exists even with decreasing payoff function of an agent as long as the function is separable between her and other agents' data contribution. We also show that a NE need not exist even with linear cost functions if the payoff functions are non-concave.

**Best response dynamics.** We now turn to computation and consider a natural procedure by which agents can find a Nash equilibrium: *best response (BR) dynamics*. Agents start with some initial sample vector $\boldsymbol{s}^0$. In each step $t$ of the BR dynamics, every agent $i$ updates their sample contribution proportional to the gradient $\frac{\partial u_i}{\partial s_i}$ in the direction of increasing utility. Concretely, for a scalar step size $\delta^t > 0$, the updates take the following form:

$$\boldsymbol{s}^{t+1} = \boldsymbol{s}^t + \delta^t \cdot g(\boldsymbol{s}^t, \boldsymbol{\mu}^t), \tag{4}$$

where $g(\boldsymbol{s}^t, \boldsymbol{\mu}^t)_i = \frac{\partial u_i(\boldsymbol{s}^t)}{\partial s_i} + \mu_i^t$ and $\boldsymbol{\mu}^t$ is chosen so that the updated sample vector $\boldsymbol{s}^{t+1}$ lies in the feasible region $\mathcal{S}$. Specifically, $\boldsymbol{\mu}^t \in \arg\min_{\boldsymbol{\mu} \in K} \|g(\boldsymbol{s}^t, \boldsymbol{\mu})\|_2$, where $K = \{\boldsymbol{\mu} : \boldsymbol{s}^t + \delta^t \cdot g(\boldsymbol{s}^t, \boldsymbol{\mu}) \in \mathcal{S}\}$. For instance, if $0 \leq s_i \leq \tau_i$, then:

$$\mu_i^t = \begin{cases} -\frac{\partial u_i}{\partial s_i}, & \text{if } s_i = 0 \text{ and } \frac{\partial u_i}{\partial s_i} < 0, \text{ or } s_i = \tau_i \text{ and } \frac{\partial u_i}{\partial s_i} > 0 \\ 0, & \text{otherwise.} \end{cases}$$

We measure convergence of the above dynamics to a Nash equilibrium via the $L^2$ norm of the update direction $g(\boldsymbol{s}^t, \boldsymbol{\mu}^t)$. Under mild assumptions on the utility functions, we show the dynamics (4) converges to an approximate Nash equilibrium where $\|g(\boldsymbol{s}^t, \boldsymbol{\mu}^t)\|_2 < \varepsilon$:

**Theorem 3.2.** *Let $G(\boldsymbol{s})$ be the Jacobian of $\boldsymbol{u} : \mathcal{S} \to \mathbb{R}^n$, i.e., $G(\boldsymbol{s})_{ij} = \frac{\partial^2 u_i(\boldsymbol{s})}{\partial s_j \partial s_i}$. Assuming agent utility functions $u_i$ satisfy*

1. *Strong concavity: $(G + \lambda \cdot I_{n \times n})$ is negative semi-definite,*

2. *Bounded derivatives: $|G_{ij}| \leq L$,*

*for constants $\lambda, L > 0$, the best response dynamics (4) with step size $\delta^t = \frac{\lambda}{n^2 L^2}$ converges to an approximate Nash equilibrium $\boldsymbol{s}^T$ where $\|g(\boldsymbol{s}^T, \boldsymbol{\mu}^T)\|_2 < \varepsilon$ in $T$ iterations, where*

$$T = \frac{2n^2 L^2}{\lambda^2} \log\left(\frac{\|g(\boldsymbol{s}^0, \boldsymbol{\mu}^0)\|_2}{\varepsilon}\right).$$

Below we sketch the proof and defer the full proof to Appendix A.

*Proof sketch.* We measure convergence of the above dynamics by the error term $\|g(\boldsymbol{s}^t, \boldsymbol{\mu}^t)\|_2$. We show the following bound:

$$\|g(\boldsymbol{s}^{t+1}, \boldsymbol{\mu}^t)\|_2^2 \leq \|g(\boldsymbol{s}^t, \boldsymbol{\mu}^t)\|_2^2 + \delta_t^2 \cdot \|G(\boldsymbol{s}')g(\boldsymbol{s}^t, \boldsymbol{\mu}^t)\|_2^2 + 2\delta_t g(\boldsymbol{s}^t, \boldsymbol{\mu}^t)^T G(\boldsymbol{s}')g(\boldsymbol{s}^t, \boldsymbol{\mu}^t).$$

We then use strong concavity to show $g(\boldsymbol{s}^t, \boldsymbol{\mu}^t)^T G(\boldsymbol{s}')g(\boldsymbol{s}^t, \boldsymbol{\mu}^t) \leq -\lambda\|g(\boldsymbol{s}^t, \boldsymbol{\mu}^t)\|_2^2$, and the bounded derivatives property to show $\|G(\boldsymbol{s}')g(\boldsymbol{s}^t, \boldsymbol{\mu}^t)\|_2 \leq nL\|g(\boldsymbol{s}^t, \boldsymbol{\mu}^t)\|_2$. For our choice of the step size $\delta^t = \frac{\lambda}{n^2 L^2}$, we can relate the error in subsequent iterations as follows:

$$\|g(\boldsymbol{s}^{t+1}, \boldsymbol{\mu}^{t+1})\|_2^2 \leq \left(1 - \frac{\lambda^2}{n^2 L^2}\right) \cdot \|g(\boldsymbol{s}^t, \boldsymbol{\mu}^t)\|_2^2.$$

This then allows us to conclude that after $T = \frac{2n^2 L^2}{\lambda^2} \log\left(\frac{\|g(s^0, \mu^0)\|_2}{\varepsilon}\right)$ iterations, we will have an approximate Nash equilibrium $s^T$ with $\|g(s^T, \mu^T)\|_2 \leq \varepsilon$. $\qquad\square$

**Implications.** The above theorem implies that agents playing the natural best response update strategy will converge quickly (in $O(\log(\varepsilon^{-1}))$ iterations) to a NE. We note that our theorem applies to the payoff functions defined by generalization bounds (eq. (2)) and observed in practice (eq. (3)) as they are strongly concave and have bounded derivatives. Moreover, under the assumptions of Theorem 3.2, our proof also implies the fast convergence of the best response dynamics in the budget-balanced mechanism we discuss in the next section.

# 4   Welfare Maximization: A Budget-Balanced Mechanism

We first note through an example that Nash equilibria need not be welfare maximizing.

*Example* 1. Consider two agents with identical payoff functions $a(s) = 8 - (s_1 + s_2)^{-1}$, and linear cost functions given by $c_1(s_1) = 5c_1$ and $c_2(s_2) = 4c_2$. The NE is given by $s^* = (0, 0.5)$, i.e., agent 1 does not contribute any data samples. The Nash welfare (which is the $p$-mean welfare in $p \to 0$ limiting case) of this NE is $u_1(s^*) \cdot u_2(s^*) = 24$. However consider another sample vector $s' = (0.2, 0.4)$ where agent 1 increases her contribution and agent 2 reduces her contribution. Then $s'$ has a higher Nash welfare of $u_1(s') \cdot u_2(s') = 25.25 > 24$.

To address the issue of NE not being welfare-maximizing, by designing a budget-balanced mechanism for agents with linear cost functions. Our mechanism is inspired from a mechanism for the efficient provisioning public goods (Falkinger et al. [2000]). We show that our mechanism always admits a NE. Moreover, when the sample vector at NE is positive, i.e. every agent participates by contributing data, the NE maximizes the $p$-mean welfare among all positive sample vectors. For the FL setting, assuming positive sample vectors is natural since the only way for an agent to participate is by making some positive data contribution.

We present our result for a more general model than one discussed so far. Not only does this generalization strengthen our result, it also naturally allows expressing agent utilities in terms of the taxes/subsidies they receive from our mechanism. In this generalization, each agent $i$ has a budget $B_i$ of which $b_i$ is unspent and the remaining is used towards the cost of sampling $s_i$ data points, i.e., $b_i + c_i(s_i) = B_i$. We assume agents have arbitrary continuous, concave utility functions of the form $u_i(b_i, \|s\|_1)$. Agents have linear cost functions $c_i(s_i) = c_i \cdot s_i + d_i$, with $c_i > 0$. This model already captures some previously discussed settings (Sec 2.1) as follows. For e.g., when payoff functions are derived from generalization bounds (Mohri et al. [2018]) or empirical evidence (Kaplan et al. [2020]), they take the form $a_i(s) = \hat{a}_i(\|s\|_1)$, for some function $\hat{a}_i$ which depends on $\|s\|_1$. Then for $u_i(b_i, \|s\|_1) = b_i + \hat{a}(\|s\|_1)$, the budget constraint implies the utility takes the form $\hat{a}_i(\|s\|_1) - c_i(s_i) + B_i$, which is the same as the utility under the original model with a constant offset. We also assume that for all $i$, $\frac{\partial u_i(b_i, S)}{\partial S} > 0$ and $\frac{\partial u_i(b_i, S)}{\partial b_i} \neq 0$.

**Mechanism $\mathcal{M}_\beta$.** We design a mechanism parametrized by a scalar $\beta \in (0, 1)$. It uses the following payment scheme. At a data contribution vector $s$, each agent $i$ is rewarded an amount

$$p_i(s) = \beta\left(c_i(s_i) - \frac{1}{n-1}\sum_{j \neq i} c_j(s_j)\right).$$

Thus if an agent incurs higher (resp. lower) sampling cost than the average cost borne by other agents, we compensate (resp. penalize) her by a subsidy (resp. tax) of $\beta$ times her excess cost. By design, our mechanism is budget-balanced: at any sample vector $s$, we have:

$$\sum_i p_i(s) = \beta\left(\sum_i c_i(s_i) - \frac{1}{n-1}\sum_{j \neq i} c_j(s_j)\right) = 0.$$

Under this mechanism, given a vector of contributions $s_{-i}$ of agents other than agent $i$, the best response of agent $i$ is any solution to the following optimization problem:

$$\max \quad u_i(b_i, s_i + \|s_{-i}\|_1)$$

$$\text{s.t.} \quad \forall i : \ b_i + (1 - \beta)c_i(s_i) + \frac{\beta}{n-1}\sum_{j \neq i} c_j(s_j) = B_i \tag{5}$$

$$\forall i : \ b_i \geq 0$$

We next define a $\beta$ value that plays a crucial role in our mechanism.

**Definition 2.** (Optimal parameter $\beta^*$) Let $A := (\sum_i c_i^{-1})^{-1}$ and $C := \sum_i c_i$. Then we define $\beta^*$ as the solution to the following equation.

$$C\beta^2 - (An(n-2) + C)\beta + A(n-1)^2 = 0, \tag{6}$$

which satisfies $0 \leq \beta^* \leq 1 - 1/n$.

The next lemma shows that (6) indeed has such a root – the proof is deferred to Appendix B.

**Lemma 1.** *The equation $C\beta^2 - (An(n-2) + C)\beta + A(n-1)^2 = 0$ of (6) has a real root $\beta^*$ where $0 \leq \beta^* \leq 1 - 1/n$.*

We now state the main result of this section. We show that for every $\beta \in [0, 1]$, a Nash equilibrium of $\mathcal{M}_\beta$ exists. Additionally, for a specific value of $\beta = \beta^*$ (Definition 2), our mechanism admits Nash equilibria which are socially efficient: it maximizes the $p$-mean welfare: $W_p(\boldsymbol{b}, \boldsymbol{s}) = (\sum_i u_i(b_i, \|s\|_1)^p)^{1/p}$, for $p \leq 1$ among all positive sample vectors $\boldsymbol{s}$.

**Theorem 4.1.** *For each $\beta \in [0, 1]$, the mechanism $\mathcal{M}_\beta$ admits a Nash equilibrium. For $\beta = \beta^*$ (Definition 2), whenever the NE $\boldsymbol{s}^*$ of $\mathcal{M}_{\beta^*}$ satisfies $\boldsymbol{s}^* > 0$, the NE $\boldsymbol{s}^*$ maximizes the $p$-mean welfare among all vectors $\boldsymbol{s} > 0$, for any $p \leq 1$.*

We now sketch the proof of the above theorem. The full proof is deferred to Appendix B.

*Proof sketch.* When $0 \leq \beta \leq 1$, the first constraint in the above program is a convex constraint even for general convex cost functions. Since $u_i(\cdot)$ is concave, a proof similar to the proof of Theorem 3.1 shows the existence of a Nash equilibrium.

The program maximizing $p$-mean welfare as follows.

$$\max \quad W_p(\boldsymbol{b}, \boldsymbol{s}) := (\sum_i u_i(b_i, \|s\|_1)^p)^{1/p}$$

$$\text{s.t.} \quad \forall i : \ b_i + (1 - \beta)c_i(s_i) + \frac{\beta}{n-1}\sum_{j \neq i} c_j(s_j) = B_i \tag{7}$$

$$\forall i : \ b_i \geq 0$$

We first show that the above program is convex. With $\mu_i$ and $\gamma_i$ as the dual variables to the first and second constraints respectively, we write down the KKT conditions of program (7) with $s_i > 0$. We then use the KKT conditions satisfied by a NE $(\boldsymbol{b}^*, \boldsymbol{s}^*)$ to find values of the dual variables $\boldsymbol{\mu}^*$ and $\boldsymbol{\gamma}^*$ so that $(\boldsymbol{b}^*, \boldsymbol{s}^*, \boldsymbol{\mu}^*, \boldsymbol{\gamma}^*)$ satisfy the KKT conditions of (7). Since KKT conditions are sufficient for optimality, this shows that the NE $(\boldsymbol{b}^*, \boldsymbol{s}^*)$ also maximizes $p$-mean welfare. $\square$

**Implications.** Theorem 4.1 shows that by augmenting the federated learning with a simple payment protocol that is budget-balanced, one can obtain NE that maximize any $p$-mean welfare function of the net-utilities of the agents. Furthermore, note that the payment augmented utility function is still essentially of the form $u_i(\boldsymbol{s}) = \hat{a}_i(\boldsymbol{s}) - (1 - \beta)c_i(s_i) - \frac{\beta}{n-1}c_j(s_j)$, with a constant offset. Since $\beta \in [0, 1]$, when $a_i(\cdot)$ are $\lambda$-strongly concave and $c_i(\cdot)$ are $\lambda$-strongly convex, the functions $u_i(\cdot)$ are $\lambda$-strongly concave. Therefore Theorem 3.2 applies, which ensures that the welfare maximizing NE can be reached through the simple best response dynamics quickly. Finally, we remark that when cost functions are identical, the value of the optimal parameter $\beta^*$ is $(1 - 1/n)$, which is exactly the value used by Falkinger et al. [2000].

*Remark* 2. Our mechanism requires that costs of the agents be publicly known in order to calculate the value of $\beta^*$ by solving (6) in Definition 2. This is a common assumption made in previous work (e.g. Karimireddy et al. [2022] and Blum et al. [2021]) and is justified in practice. Indeed, costs are common knowledge in many real-world applications e.g. (1) in many ML applications, each agent derives their training data from manually labeling a subset of a publicly available dataset like CIFAR or ImageNet, and the cost of labeling dataset is usually known; (2) in autonomous driving, where each data point is a random path taken under random external conditions.

# 5 Distributed Algorithm and Empirical Evaluation

In this section, we realize our budget-balanced mechanism in a real-world FL system. We perform the evaluation on the MNIST (LeCun et al. [2010]) and CIFAR-10 (Krizhevsky [2009]) datasets. We compare our mechanism with two baselines: the standard FedAvg and MWFed Blum et al. [2021]. We denote the vanilla mechanism without budget balancing as FedBR and the budget-balanced mechanism as FedBR-BG. We demonstrate that compared to the FedBR, FedBR-BG achieves better $p$-mean welfare for $p \leq 1$. We also show that the standard FL protocol FedAvg gives significantly lower welfare since it does not allow agents to change their contribution.

## 5.1 Algorithm Details

We first define the concrete forms for payoff and cost functions. According to standard practice in FL, an agents payoff is measured through the accuracy evaluated on her local test data given model $\theta$, which has the form $a_i(\boldsymbol{s}, \theta) = \frac{1}{|D_i^*|} \sum_{(x,y) \in D_i^*} \mathbb{1}[\theta(x) = y]$, where $D_i^*$ is the test data for agent $i$. We note that this form inherently aligns with $a_i(\boldsymbol{s}) = \hat{a}_i(\|\boldsymbol{s}\|_1)$ because the received global model $\theta$ is trained on $\|\boldsymbol{s}\|_1$ samples. We consider linear cost functions where $c_i(s_i) = c_i s_i$.

Now we derive the update rule for agent contributions for FedBR-BG based on best-response dynamics. We rewrite the utility of agent $i$ in the budget-balancing mechanism as $u_i(\boldsymbol{s}) = \hat{a}_i(\sum_i s_i) - (1 - \beta)c_i s_i - \frac{\beta}{n-1} \sum_{j \neq i} c_j s_j$. We then compute its gradient with respect to $s_i$: $\frac{\partial u_i}{\partial s_i} = \frac{\partial \hat{a}_i(\sum_i s_i)}{\partial s_i} - (1 - \beta)c_i$. Since the accuracy function is generally unknown in practice, the server can train a public accuracy function $\tilde{a} : \tilde{\mathcal{S}} \to \mathbb{R}_{\geq 0}$ and broadcast it to all agents. We obtain an estimate of the accuracy $\tilde{a}$ by evaluating models trained on different numbers of samples, in increasing intervals of $\Delta s$. For example, if the server trains models on $0, 2, 4, \cdots$ samples and evaluates their accuracy to obtain the estimated accuracy $\tilde{a}$, the interval $\Delta s = 2$ in this case. We assume $(\tau_1, \ldots, \tau_n) \in \tilde{\mathcal{S}}$ as the server can be a service provider with plenty of data sources. With $\tilde{a}$, we can approximate the gradient as $\frac{\partial u_i}{\partial s_i} \approx \frac{\tilde{a}(\sum_i s_i + \Delta s) - \tilde{a}(\sum_i s_i)}{\Delta s} - (1 - \beta)c_i$. Similarly, $\frac{\partial u_i}{\partial s_i} \approx \frac{\tilde{a}(\sum_i s_i + \Delta s) - \tilde{a}(\sum_i s_i)}{\Delta s} - c_i$ for FedBR, and $u_i(\boldsymbol{s}) = a_i(\sum_i s_i) - c_i s_i$ for FedAvg, MWFed and FedBR.

We further assume that $s_i$ is a multiple of $\Delta s$ to ensure that $\tilde{a}$ is always well-defined for our chosen contributions. Finally, we leverage best-response dynamics to update agent contributions. We present the full description of the algorithm for FedBR-BG and FedBR as Algorithm 1 and Algorithm 2 in Appendix C, respectively.

## 5.2 Experiment Setup

We conduct the experiments with 10 agents for MNIST and 100 agents for CIFAR-10. For MNIST, we use a CNN as the global model, which has two $5 \times 5$ convolution layers followed by two fully connected layers with ReLU activation. For CIFAR-10, we use VGG11 (Simonyan and Zisserman [2014]). We consider the *i.i.d.* setting, i.e., the local data of agents are sampled from the same distribution. For both datasets, each agent has 100 training images and 10 testing images, i.e., $\tau_i = 100, \forall i \in [n]$.

We randomly initialize the contributions as a multiplier of 10 in $[0, 100]$. In each contribution updating step, we re-initialize the global model and perform FedAvg for 50 communication rounds. We set global learning rate $\eta$ to 1.0, local learning rate $\alpha$ to 0.01, and momentum to 0.9. We set the number of contribution updating steps to 100 and the sample number interval to 10. For evaluating FedAvg, we simply optimize the global model with the same hyperparameters while keeping individual contribution to $\tau_i$.

## 5.3 Experiment Results

We show the $p$-mean welfare of our method and baselines on MNIST and CIFAR-10 in Table 1. We observe that FedBR-BG achieves better $p$-mean welfare on both datasets compared to FedBR and FedAvg, verifying our theoretical results. Note that the $p$-mean of FedAvg is significantly lower since agents always contribute all their data in FedAvg, which incurs a high cost so that the marginal gain becomes limited.

Table 1: $p$-mean welfare of our budget-balanced mechanism FedBR-BG and baselines on MNIST and CIFAR-10. We report the results for different $p$. The cost for adding one data sample $c_i$ is 0.005 for every agent.

| Method | MNIST | | | | | CIFAR-10 | | | | |
|--------|-------|---|---|---|---|----------|---|---|---|---|
| | $p = 0.2$ | $p = 0.4$ | $p = 0.6$ | $p = 0.8$ | $p = 1.0$ | $p = 0.2$ | $p = 0.4$ | $p = 0.6$ | $p = 0.8$ | $p = 1.0$ |
| FedAvg | 48985.23 | 154.99 | 22.763 | 8.726 | 4.909 | 42386.21 | 135.92 | 23.528 | 8.381 | 4.582 |
| MWFed | 51326.49 | 158.92 | 21.648 | 8.803 | 5.230 | 48178.29 | 142.91 | 23.981 | 8.879 | 4.891 |
| FedBR | 53395.21 | 168.85 | 24.784 | 9.495 | 5.340 | 58297.23 | 178.32 | 26.187 | 9.675 | 5.681 |
| FedBR-BG | **54589.31** | **172.63** | **25.340** | **9.708** | **5.459** | **60385.32** | **183.23** | **27.958** | **9.981** | **5.891** |

## 6 Discussion

In this paper, we formulated a federated learning framework which incorporates both payoffs an agent receives from sharing data as well as the cost she incurs due to sharing data. We show the existence of Nash equilibria under the assumption of concave payoffs and convex costs, which are mild assumptions observed in practice. We then note that while NE exist, they may not maximize any notion of welfare for the agents, leaving agents with less incentive to participate. We address this issue via a budget-balanced mechanism with payments whose NE maximize the $p$-mean welfare of the agent utilities. Our experiments show that FedBR-BG achieves better $p$-mean welfare compared to FedBR and FedAvg, verifying our theoretical results.

We conclude by discussion some directions for future work. Our mechanism required that costs of the agents be publicly known, or at least verifiable. Studying incentives when costs are not common knowledge is an interesting question. Another assumption of our mechanism was that an agent's payoff depends on the number of data samples. Designing welfare-maximizing mechanisms for general settings where payoff functions take more general forms is another direction for future work.

**Acknowledgements.** This work is partially supported by the National Science Foundation under grant No. 1750436, No. 1910100, No. 2046726, No. 2229876, DARPA GARD, the National Aeronautics and Space Administration (NASA) under grant no. 80NSSC20M0229, the Alfred P. Sloan Fellowship, and the Amazon research award.

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

# A    Appendix to Section 3

We first show that a Nash equilibrium exists when agent payoff functions are *separable*, i.e., for every agent $i$ there are functions $g_i : S_i \to \mathbb{R}_{\geq 0}$ and $h_i : \bigtimes_{j \neq i} S_j \to \mathbb{R}_{\geq 0}$ s.t. for all $\boldsymbol{s} \in \mathcal{S}$, $a_i(\boldsymbol{s}) = g_i(s_i) + h_i(\boldsymbol{s}_{-i})$.

**Theorem A.1.** *In any federated learning problem where agent payoff functions are separable, a Nash equilibrium exists.*

*Proof.* When the payoff function of an agent $i$ is separable, the best response to any contribution vector $\boldsymbol{s}_{-i}$ is independent of $\boldsymbol{s}_{-i}$:

$$f_i(\boldsymbol{s}_{-i}) = \arg\max_{x \in S_i} a_i(x, \boldsymbol{s}_{-i}) - c_i(x) = \arg\max_{x \in S_i} g_i(x) + h_i(\boldsymbol{s}_{-i}) - c_i(x)$$
$$= \arg\max_{x \in S_i} g_i(x) - c_i(x). \qquad \text{(since } h_i(\boldsymbol{s}_{-i}) \text{ is independent of } x\text{)}$$

Let $F_i := \arg\max_{x \in S_i} g_i(x) - c_i(x)$. Clearly $F_i \neq \emptyset$ since $S_i \neq \emptyset$. Then any $\boldsymbol{s} \in \bigtimes_i F_i$ satisfies $\boldsymbol{s} \in f(\boldsymbol{s})$ by definition. By Proposition 1 any such sample vector is a Nash equilibrium. $\qquad\square$

Next, we present a negative result showing that there are federated learning settings where a Nash equilibrium is not guaranteed to exist.

**Theorem A.2.** *There exists a federated learning problem in which a Nash equilibrium does not exist. Moreover, the instance has three agents with continuous, non-decreasing, non-concave payoff functions and linear cost functions.*

*Proof.* Let $\varepsilon \in (0, \frac{1}{16})$. Let $e : [0, 1] \to [0, 1]$ be a function given by:

$$e(x) = \begin{cases} 0, & \text{if } 0 \leq x \leq \frac{1}{2} - \varepsilon, \\ \frac{1}{2} + \frac{1}{2\varepsilon}(x - \frac{1}{2}), & \text{if } \frac{1}{2} - \varepsilon \leq x \leq \frac{1}{2} + \varepsilon, \\ 1, & \text{if } \frac{1}{2} + \varepsilon \leq x \leq 1. \end{cases} \tag{8}$$

Essentially the function $e$ is a continuous, piece-wise linear function connecting $(0, 0)$, $(\frac{1}{2} - \varepsilon, 0)$, $(\frac{1}{2} + \varepsilon, 1)$ and $(1, 1)$.

Now consider the following federated learning instance with $n = 3$ agents, where $S_1 = S_2 = S_3 = [0, 1]$. The payoff functions are given by:

$$\begin{aligned} a_1(\boldsymbol{s}) &= e(s_1) + e(s_3) - e(s_1) \cdot e(s_3) \\ a_2(\boldsymbol{s}) &= e(s_2) + e(s_1) - e(s_2) \cdot e(s_1) \\ a_3(\boldsymbol{s}) &= e(s_3) + e(s_2) - e(s_3) \cdot e(s_2), \end{aligned} \tag{9}$$

and the cost functions are $c_i(s_i) = \frac{1}{4} s_i$ for all $i \in [3]$. Notice that the payoff functions are increasing in $s_j$ for every $j \in [3]$ and are continuous since $e$ is continuous.

We now show that this instance does not admit a Nash equilibrium. Let us first evaluate the best response set $f_1(s_2, s_3)$. Note that $u_1(\boldsymbol{s}) = e(s_1) \cdot (1 - e(s_3)) + e(s_3) - \frac{1}{4} s_1$. Since $u_1(\boldsymbol{s})$ is independent of $s_2$, $f_1(s_2, s_3)$ only depends on $s_3$.

- Case 1. $s_3 \leq \frac{1}{2} - \varepsilon$. Then $u_1(\boldsymbol{s}) = e(s_1) - \frac{1}{4} s_1$, which is maximized at $s_1 = \frac{1}{2} + \varepsilon$ and results in a utility of $\frac{7}{8} - \frac{\varepsilon}{4}$.

- Case 2. $s_3 \geq \frac{1}{2} + \varepsilon$. Then $u_1(\boldsymbol{s}) = 1 - \frac{1}{4} s_1$, which is maximized at $s_1 = 0$ and results in a utility of 1.

- Case 3. $\frac{1}{2} - \varepsilon \leq s_3 \leq \frac{1}{2} + \varepsilon$. We consider the intervals in which the best response $s_1$ to such an $s_3$ can lie:

    - $s_1 \leq \frac{1}{2} - \varepsilon$. In this range, $u_1(\boldsymbol{s}) = e(s_3) - \frac{1}{4} s_1$, which is maximized at $s_1 = 0$ and results in a utility of $e(s_3)$.

- $s_1 \geq \frac{1}{2} + \varepsilon$. In this range, $u_1(\boldsymbol{s}) = 1 - \frac{1}{4}s_1$, which is maximized at $s_1 = \frac{1}{2} + \varepsilon$ and results in a utility of $\frac{7}{8} - \frac{\varepsilon}{4}$.

- $\frac{1}{2} - \varepsilon \leq s_1 \leq \frac{1}{2} + \varepsilon$. In this range, using the definition of $e(s_1)$ (eq. 8) we obtain:

$$u_1(\boldsymbol{s}) = \left( \frac{1 - e(s_3)}{2\varepsilon} - \frac{1}{4} \right) \cdot s_1 + (1 - e(s_3)) \cdot \left( \frac{1}{2} - \frac{1}{4\varepsilon} \right) + e(s_3).$$

Thus $u_1(\boldsymbol{s})$ is a linear function in $s_1$ with slope $\frac{1-e(s_3)}{2\varepsilon} - \frac{1}{4}$. If the slope is positive, then the best response in the current interval is $s_1 = \frac{1}{2} + \varepsilon$, and gives a utility of $\frac{7}{8} - \frac{\varepsilon}{4}$. If the slope is negative, then $s_1 = \frac{1}{2} - \varepsilon$ is the best response in the current interval and gives a utility of $e(s_3) - \frac{1}{4}(\frac{1}{2} - \varepsilon)$. However $s_1 = 0$ gives a utility of $e(s_3)$ implying that $s_1 = \frac{1}{2} - \varepsilon$ cannot be a best response. Finally if the slope is zero, then it must mean that $e(s_3) = 1 - \frac{\varepsilon}{2}$, and the utility is $\frac{\varepsilon}{2}(\frac{1}{2} - \frac{1}{4\varepsilon}) + 1 - \frac{\varepsilon}{2} = \frac{7}{8} - \frac{\varepsilon}{4}$. However responding with $s_1 = 0$ gives a utility of $e(s_3) = 1 - \frac{\varepsilon}{2}$, which exceeds $\frac{7}{8} - \frac{\varepsilon}{4}$, since $\varepsilon < \frac{1}{16}$. Thus, the best response does not lie in $(\frac{1}{2} - \varepsilon, \frac{1}{2} + \varepsilon)$ and $s_1 = 0$ is the overall best response.

The above discussion shows that the best response $f_1(s_2, s_3) \subseteq \{0, \frac{1}{2} + \varepsilon\}$. By symmetry, the same holds for $f_2$ and $f_3$. Suppose there exists a Nash equilibrium $\boldsymbol{s}^* = (s_1^*, s_2^*, s_3^*)$. By Proposition 1, $\boldsymbol{s}^* \in f(\boldsymbol{s}^*)$. Since the above discussion implies $s_3^* \in \{0, \frac{1}{2} + \varepsilon\}$, we consider two cases:

- Suppose $s_3^* = 0$. Then

$$
\begin{aligned}
s_3^* = 0 &\implies s_1^* = \frac{1}{2} + \varepsilon && \text{(Case 1 for agent 1)} \\
&\implies s_2^* = 0 && \text{(Case 2 for agent 2)} \\
&\implies s_3^* = \frac{1}{2} + \varepsilon, && \text{(Case 1 for agent 3)}
\end{aligned}
$$

which is a contradiction.

- Suppose $s_3^* = \frac{1}{2} + \varepsilon$. Then

$$
\begin{aligned}
s_3^* = \frac{1}{2} + \varepsilon &\implies s_1^* = 0 && \text{(Case 2 for agent 1)} \\
&\implies s_2^* = \frac{1}{2} + \varepsilon && \text{(Case 1 for agent 2)} \\
&\implies s_3^* = 0, && \text{(Case 2 for agent 3)}
\end{aligned}
$$

which is also a contradiction.

This shows that there is no $\boldsymbol{s}^*$ such that $\boldsymbol{s}^* \in f(\boldsymbol{s}^*)$, implying that the above instance does not admit a Nash equilibrium. $\qquad\square$

We now prove the fast convergence of best response dynamics.

**Theorem 3.2.** *Let $G(\boldsymbol{s})$ be the Jacobian of $\boldsymbol{u} : \mathcal{S} \to \mathbb{R}^n$, i.e., $G(\boldsymbol{s})_{ij} = \frac{\partial^2 u_i(\boldsymbol{s})}{\partial s_j \partial s_i}$. Assuming agent utility functions $u_i$ satisfy*

1. *Strong concavity: $(G + \lambda \cdot I_{n \times n})$ is negative semi-definite,*

2. *Bounded derivatives: $|G_{ij}| \leq L$,*

*for constants $\lambda, L > 0$, the best response dynamics (4) with step size $\delta^t = \frac{\lambda}{n^2 L^2}$ converges to an approximate Nash equilibrium $\boldsymbol{s}^T$ where $\|g(\boldsymbol{s}^T, \boldsymbol{\mu}^T)\|_2 < \varepsilon$ in $T$ iterations, where*

$$T = \frac{2n^2 L^2}{\lambda^2} \log \left( \frac{\|g(\boldsymbol{s}^0, \boldsymbol{\mu}^0)\|_2}{\varepsilon} \right).$$

*Proof.* Observe that $\boldsymbol{\mu}^t$ is chosen s.t. $\|g(\boldsymbol{s}^t, \boldsymbol{\mu}^t)\|_2$ is minimized among all $\mu$ s.t. the updated sample vector $\boldsymbol{s}^{t+1}$ remains in $\mathcal{S}$. Thus:

$$\|g(\boldsymbol{s}^{t+1}, \boldsymbol{\mu}^{t+1})\|_2 \leq \|g(\boldsymbol{s}^{t+1}, \boldsymbol{\mu}^t)\|_2 \tag{10}$$

Using Taylor's expansion, we have:

$$g(\boldsymbol{s}^{t+1}, \boldsymbol{\mu}^t) = g(\boldsymbol{s}^t, \boldsymbol{\mu}^t) + H(\boldsymbol{s}', \mu^t) \cdot (\boldsymbol{s}^{t+1} - \boldsymbol{s}^t),$$

where $H_{ij}(\boldsymbol{s}', \boldsymbol{\mu}^t) = \frac{\partial g(\boldsymbol{s}', \boldsymbol{\mu}^t)}{\partial s_j}$, and $\boldsymbol{s}' = \boldsymbol{s}^t + \alpha(\boldsymbol{s}^{t+1} - \boldsymbol{s}^t)$ for some $\alpha \in [0, 1]$.

By definition, $g(\boldsymbol{s}^t, \mu^t)_i = \frac{\partial u_i(\boldsymbol{s}^t)}{\partial s_i} + \mu_i^t$. Thus $H_{ij}(\boldsymbol{s}', \boldsymbol{\mu}^t) = \frac{\partial^2 u_i(\boldsymbol{s}^t)}{\partial s_j \partial s_i} = G_{ij}(\boldsymbol{s}')$, hence $H(\boldsymbol{s}', \boldsymbol{\mu}^t) = G(\boldsymbol{s}')$. The BR dynamics update rule (4) implies $\boldsymbol{s}^{t+1} - \boldsymbol{s}^t = \delta^t \cdot g(\boldsymbol{s}^t, \boldsymbol{\mu}^t)$. We therefore have $g(\boldsymbol{s}^{t+1}, \boldsymbol{\mu}^t) = (I_{n \times n} + \delta^t \cdot G(\boldsymbol{s}')) \cdot g(\boldsymbol{s}^t, \boldsymbol{\mu}^t)$. Taking the $L^2$ norm, we get:

$$\|g(\boldsymbol{s}^{t+1}, \boldsymbol{\mu}^t)\|_2^2 = \|g(\boldsymbol{s}^t, \boldsymbol{\mu}^t)\|_2^2 + \delta_t^2 \cdot \|G(\boldsymbol{s}')g(\boldsymbol{s}^t, \boldsymbol{\mu}^t)\|_2^2 + 2\delta_t g(\boldsymbol{s}^t, \boldsymbol{\mu}^t)^T G(\boldsymbol{s}')g(\boldsymbol{s}^t, \boldsymbol{\mu}^t), \tag{11}$$

By the strong concavity assumption, for a constant $\lambda > 0$, $G + \lambda \cdot I_{n \times n}$ is negative semi-definite, i.e., $v^T(G + \lambda \cdot I_{n \times n})v \leq 0$ for any $v \in \mathbb{R}^n$. With $v = g(\boldsymbol{s}^t, \boldsymbol{\mu}^t)$, we have:

$$g(\boldsymbol{s}^t, \boldsymbol{\mu}^t)^T G(\boldsymbol{s}')g(\boldsymbol{s}^t, \boldsymbol{\mu}^t) \leq -\lambda \cdot \|g(\boldsymbol{s}^t, \boldsymbol{\mu}^t)\|_2^2. \tag{12}$$

Next we use the fact that the $L^2$ norm $\|A\|_2$ of an $n \times n$ matrix $A$ is bounded by its Frobenius norm $\|A\|_F$:

$$\|A\|_2 := \sup_{x \neq 0} \frac{\|Ax\|_2}{\|x\|_2} \leq \|A\|_F := \sqrt{\sum_i \sum_j |A_{ij}|^2}$$

By the bounded derivatives assumption, we have $|G(\boldsymbol{s}')_{ij}| \leq L$, which implies that $\|G(\boldsymbol{s}')\|_F = \sqrt{\sum_i \sum_j L^2} = nL$. This gives:

$$\|G(\boldsymbol{s}')g(\boldsymbol{s}^t, \boldsymbol{\mu}^t)\|_2 \leq nL\|g(\boldsymbol{s}^t, \mu^t)\|_2. \tag{13}$$

Using (12) and (13) in (11), we get:

$$\|g(\boldsymbol{s}^{t+1}, \boldsymbol{\mu}^t)\|_2^2 = (1 + \delta_t^2 \cdot n^2 L^2 - 2\delta^t \lambda) \cdot \|g(\boldsymbol{s}^t, \boldsymbol{\mu}^t)\|_2^2,$$

Since $\delta^t = \frac{\lambda}{n^2 L^2}$, the above equation together with (10) gives:

$$\|g(\boldsymbol{s}^{t+1}, \boldsymbol{\mu}^{t+1})\|_2^2 \leq \left(1 - \frac{\lambda^2}{n^2 L^2}\right) \cdot \|g(\boldsymbol{s}^t, \boldsymbol{\mu}^t)\|_2^2.$$

Using $(1 - x)^r \leq e^{-xr}$ repeatedly we obtain that:

$$\|g(\boldsymbol{s}^t, \boldsymbol{\mu}^t)\|_2 \leq e^{-\frac{\lambda^2}{2n^2 L^2} \cdot t} \cdot \|g(\boldsymbol{s}^0, \boldsymbol{\mu}^0)\|_2.$$

Thus if we want the error $\|g(\boldsymbol{s}^t, \boldsymbol{\mu}^t)\|_2 \leq \varepsilon$, $T = \frac{2n^2 L^2}{\lambda^2} \log\left(\frac{\|g(\boldsymbol{s}^0, \boldsymbol{\mu}^0)\|_2}{\varepsilon}\right)$ iterations suffice, as claimed. $\qquad\square$

# B Appendix to Section 4

**Lemma 1.** *The equation $C\beta^2 - (An(n-2) + C)\beta + A(n-1)^2 = 0$ of (6) has a real root $\beta^*$ where $0 \leq \beta^* \leq 1 - 1/n$.*

*Proof.* Using the quadratic formula, we see that $\beta^*$ given by:

$$\beta^* = \frac{An(n-2) + C - \sqrt{(An(n-2) + C)^2 - 4AC(n-1)^2}}{2C} \tag{14}$$

We first argue $\beta^*$ is real, by showing $(An(n-2) + C)^2 - 4AC(n-1)^2 \geq 0$. This is equivalent to showing $q(y) := (y + n(n-2))^2 - 4(n-1)^2 y \geq 0$, where $y = C/A$. Expanding $q$, we have $q(y) = y^2 - 2(n^2 - 2n + 2)y + n^2(n-2)^2$. The roots of $q$ are:

$$y_1, y_2 = \frac{2(n^2 - 2n + 2) \pm \sqrt{4(n^2 - 2n + 2)^2 - 4n^2(n-2)^2}}{2} = (n^2 - 2n + 2) \pm 2(n-1),$$

i.e., $y_1 = (n-2)^2$ and $y_2 = n^2$. Since $q(y)$ has a positive leading coefficient, we have that $q(y) \geq 0$ for all $y \geq y_2 = n^2$. Thus it remains to show that $y = C/A \geq n^2$. To see this, we use the AM-HM inequality:

$$\frac{C}{n} = \frac{c_1 + \cdots + c_n}{n} \geq \frac{n}{\frac{1}{c_1} + \cdots + \frac{1}{c_n}} = \frac{n}{A}, \tag{15}$$

implying $C/A \geq n^2$ as desired. This shows that the root $\beta^*$ of equation (6) is real, hence well-defined.

We now show $0 \leq \beta^* \leq 1 - 1/n$. From (14), we see:

$$\begin{aligned}
\beta^* &= \frac{An(n-2) + C - \sqrt{(An(n-2) + C)^2 - 4AC(n-1)^2}}{2C} \\
&\geq \frac{An(n-2) + C - \sqrt{(An(n-2) + C)^2}}{2C} = 0
\end{aligned}$$

Further, from (14) we also have:

$$\begin{aligned}
\beta^* &= \frac{An(n-2) + C - \sqrt{(An(n-2) + C)^2 - 4AC(n-1)^2}}{2C} \\
&\leq \frac{An(n-2) + C}{2C} = \frac{Cn(n-2)/n^2 + C}{2C} = 1 - \frac{1}{n},
\end{aligned}$$

where we used $A/C \leq 1/n^2$ (15) in the last inequality. This concludes the proof of Lemma 1. $\square$

**Theorem 4.1.** *For each $\beta \in [0,1]$, the mechanism $\mathcal{M}_\beta$ admits a Nash equilibrium. For $\beta = \beta^*$ (Definition 2), whenever the NE $\boldsymbol{s}^*$ of $\mathcal{M}_{\beta^*}$ satisfies $\boldsymbol{s}^* > 0$, the NE $\boldsymbol{s}^*$ maximizes the $p$-mean welfare among all vectors $\boldsymbol{s} > 0$, for any $p \leq 1$.*

*Proof.* When $0 \leq \beta \leq 1$, the program (5) is a convex program for general convex cost functions. Since $u_i(\cdot)$ is concave, a proof similar to the proof of Theorem 3.1 shows the existence of a Nash equilibrium.

We now show the welfare-maximizing property. For simplicity, we only consider feasible strategies where each agent participates in the mechanism, i.e., $s_i > 0$. Let $\rho_i$ and $\lambda_i$ as the dual variables to the first and second constraints respectively for each $i$, and let $S = \|\boldsymbol{s}\|_1$. Writing the KKT conditions and eliminating all $\rho_i$, we get that a NE $(\boldsymbol{b}^*, \boldsymbol{s}^*)$ together with dual variables $\lambda^*$ satisfies:

$$\forall i: \quad \frac{\partial u_i(b_i^*, S^*)}{\partial S} = (1 - \beta) \cdot c_i \cdot \left( \frac{\partial u_i(b_i^*, S^*)}{\partial b_i} + \lambda_i^* \right) \quad \text{(from stationarity conditions)} \tag{16}$$

$$\forall i: \quad \lambda_i^* \geq 0 \qquad \text{(dual feasibility)} \tag{17}$$

$$\forall i: \quad \lambda_i^* \cdot b_i = 0 \qquad \text{(complimentary slackness)} \tag{18}$$

Now we turn to the $p$-mean welfare maximizing solution which is an optimal solution to the following program.

$$\max \quad W_p(\boldsymbol{b}, \boldsymbol{s}) := \left( \sum_i u_i(b_i, \|s\|_1)^p \right)^{1/p}$$

$$\text{s.t.} \quad \forall i: b_i + (1-\beta)c_i(s_i) + \frac{\beta}{n-1} \sum_{j \neq i} c_j(s_j) = B_i \tag{19}$$

$$\forall i: b_i \geq 0$$

The following lemma establishes that (19) is a convex program. For ease of readability we defer its proof to B.1.

**Lemma 2.** *For $\beta \in [0,1]$ and $p \leq 1$, the program (19) is convex.*

We can now write the KKT conditions of program (19). By letting $\mu_i$ and $\gamma_i$ denote the dual variables corresponding to the first and second constraints respectively for each $i$ and $S = \|\boldsymbol{s}\|_1$, the KKT conditions (considering only solutions with $s_i > 0$) are:

$$\forall i: \quad \left( \sum_j u_j^p \right)^{1/p - 1} \sum_k u_k^{p-1} \frac{\partial u_k}{\partial S} = c_i \cdot \left[ \mu_i(1 - \beta) + \frac{\beta}{n-1} \sum_{k \neq i} \mu_k \right] \quad \text{(stationarity)} \tag{20}$$

$$\forall i: \quad (\sum_j u_j^p)^{1/p-1} u_i^{p-1} \frac{\partial u_i}{\partial b_i} = \mu_i - \gamma_i \qquad \text{(stationarity)} \qquad (21)$$

$$\forall i: \quad \gamma_i \geq 0 \qquad \text{(dual feasibility)} \qquad (22)$$

$$\forall i: \quad \gamma_i \cdot b_i = 0 \qquad \text{(complimentary slackness)} \qquad (23)$$

Since KKT conditions are sufficient for optimality, to prove Theorem 4.1 it suffices to show that for an NE $(\boldsymbol{b}^*, \boldsymbol{s}^*)$, there exist dual variables $\boldsymbol{\mu}^*$ and $\boldsymbol{\gamma}^*$ which satisfy (20)-(23) for $\beta = \beta^*$.

Let $\alpha := (\sum_j u_j(b_j^*, \boldsymbol{s}^*)^p)^{1/p-1} \sum_k u_k(b_j^*, \boldsymbol{s}^*)^{p-1} \frac{\partial u_k(b_k^*, \boldsymbol{s}^*)}{\partial S}$, i.e., the common value of the equality (20) at the NE $(\boldsymbol{b}^*, \boldsymbol{s}^*)$. The equation (20) then becomes $\alpha \cdot c_i^{-1} = \mu_i(1-\beta) + \frac{\beta}{n-1} \sum_{k \neq i} \mu_k$. Summing these over all $i$ and letting $T = \sum_j \mu_j$, we obtain:

$$\alpha \cdot (\sum_i c_i^{-1}) = \sum_i [\mu_i(1-\beta) + \frac{\beta}{n-1} \sum_{k \neq i} \mu_k] = T.$$

Putting this back in (20), we obtain the following expression for $\mu_i^*$, which can be computed from the NE $(\boldsymbol{b}^*, \boldsymbol{s}^*)$ with $T = \alpha \cdot (\sum_i c_i^{-1})$:

$$\mu_i^* = \frac{\frac{T c_i^{-1}}{\sum_i c_i^{-1}} - \frac{\beta T}{n-1}}{1 - \frac{\beta n}{n-1}}. \qquad (24)$$

Recall that the NE $(\boldsymbol{b}^*, \boldsymbol{s}^*)$ satisfies (16)-(18) for some dual variables $\lambda^*$. We define $\gamma_i^*$ as follows:

$$\gamma_i^* = \mu_i^* \cdot \left( \frac{\lambda_i^*}{\lambda_i^* + \frac{\partial u_i(b_i^*, \boldsymbol{s})}{\partial b_i}} \right) \qquad (25)$$

The next lemma proves Theorem 4.1.

**Lemma 3.** *A NE $(\boldsymbol{b}^*, \boldsymbol{s}^*)$ with $\boldsymbol{\mu}^*$ and $\boldsymbol{\gamma}^*$ defined by* (24) *and* (25) *satisfy the KKT conditions* (20)-(23) *of program* (19).

*Proof.* First observe that at the NE, $(1-\beta)c_i \cdot \left( \frac{\partial u_i(b_i^*, S^*)}{\partial b_i} + \lambda_i^* \right) = \frac{\partial u_i(b_i^*, S^*)}{\partial S} > 0$ by assumption. Since $\beta \in (0,1)$ and $c_i > 0$, we have $\frac{\partial u_i(b_i^*, S^*)}{\partial b_i} + \lambda_i^* > 0$. Together with $\lambda_i^* \geq 0$ (17), this shows $\gamma_i^* \geq 0$ thus satisfying dual feasibility (22).

Next we show complimentary slackness (23) holds. For any $i$, $\lambda_i^* \cdot b_i = 0$ due to (18). Then by the definition of $\gamma_i^*$, we have $\gamma_i^* \cdot b_i = 0$ for all $i$.

Finally, we show that equations (20) and (21) are satisfied for a specific choice of $\beta = \beta^*$. Together, (20) and (21) imply that an optimal solution to program (19) satisfies:

$$\forall i: \quad \sum_k (\mu_k - \gamma_k) \cdot \frac{\partial u_k / \partial S}{\partial u_k / \partial b_k} = c_i \cdot [\mu_i(1-\beta) + \frac{\beta}{n-1} \sum_{k \neq i} \mu_k] \qquad (26)$$

The choice of $\gamma_i^*$ from equation 25 implies that $\mu_i^* - \gamma_i^* = \mu_i^* \cdot \left( \frac{\partial u_i(b_i^*, \boldsymbol{s})/\partial b_i}{\partial u_i(b_i^*, \boldsymbol{s})/\partial b_i + \lambda_i^*} \right)$. Moreover at the NE, equation (16) implies that:

$$(\mu_i^* - \gamma_i^*) \cdot \frac{\partial u_i(b_i^*, \boldsymbol{s})/\partial S}{\partial u_i(b_i^*, \boldsymbol{s})/\partial b_i} = \mu_i^* \cdot \left( \frac{\partial u_i(b_i^*, \boldsymbol{s})/\partial b_i}{\partial u_i(b_i^*, \boldsymbol{s})/\partial b_i + \lambda_i^*} \right) \cdot (1-\beta)c_i \cdot \left( 1 + \frac{\lambda_i^*}{\partial u_i(b_i^*, \boldsymbol{s})/\partial b_i} \right)$$
$$= \mu_i^* \cdot (1-\beta)c_i.$$

Using the above in (26), it only remains to be argued that $\boldsymbol{\mu}^*$, $\boldsymbol{b}^*$ and $\boldsymbol{s}^*$ satisfy:

$$\forall i: \quad (1-\beta) \cdot \sum_k \mu_k^* \cdot c_k = c_i \cdot [\mu_i^*(1-\beta) + \frac{\beta}{n-1} \sum_{k \neq i} \mu_k^*] = \alpha,$$

for $\beta = \beta^*$. By plugging in the value of $\mu_i^*$ from (24) and using $\alpha = T \cdot (\sum_k c_k^{-1})^{-1}$, we get:

$$(1 - \beta) \cdot \sum_k \left\{ \frac{Tc_k^{-1}(\sum_i c_i^{-1})^{-1} - \frac{\beta T}{n-1}}{1 - \frac{\beta n}{n-1}} \right\} \cdot c_k = T \cdot (\sum_k c_k^{-1})^{-1}.$$

Let us define $A := (\sum_i c_i^{-1})^{-1}$ and $C := \sum_i c_i$. Manipulating the above expression, the above equation then becomes:

$$C\beta^2 - (An(n-2) + C)\beta + A(n-1)^2 = 0,$$

which is true for $\beta = \beta^*$ since it is exactly the definition of $\beta^*$ (Definition 2).

Thus for $\beta = \beta^*$, the NE $(\boldsymbol{b}^*, \boldsymbol{s}^*)$ with dual variables $\boldsymbol{\mu}$ and $\boldsymbol{\gamma}$ as defined in (24) and (25) respectively satisfy the KKT conditions of program (19). $\qquad\square$

$\qquad\square$

## B.1   Proof of Lemma 2

**Lemma 2.** *For $\beta \in [0, 1]$ and $p \leq 1$, the program (19) is convex.*

*Proof.* For $\beta \in [0, 1]$ the constraints of program 19 are convex since $c_i(\cdot)$ are convex functions. It remains to be shown that the objective $W_p(\boldsymbol{b}, \boldsymbol{s}) := (\sum_i u_i(b_i, \|s\|_1)^p)^{1/p}$ to be maximized is concave.

We use the following standard fact about the concavity of composition of functions (see e.g. Boyd and Vandenberghe [2004], Page 86).

**Proposition 2.** *Let $h : \mathbb{R}^n \to \mathbb{R}$ and $g_i : \mathbb{R}^k \to \mathbb{R}$ and let $f : \mathbb{R}^n \to \mathbb{R}$ be given by $f(x) = h(g(x)) = h(g_1(x), \ldots, g_n(x))$. Then $f$ is concave if $h$ is concave, $h$ is non-decreasing in each argument and $g_i$ are concave.*

Note that $W_p(\boldsymbol{b}, \boldsymbol{s}) = h(g(\boldsymbol{b}, \boldsymbol{s}))$, where $h(x_1, \ldots, x_n) = (\sum_i x_i^p)^{1/p}$ and $g_i(\boldsymbol{b}, \boldsymbol{s}) = u_i(\boldsymbol{b}, \boldsymbol{s})$.

We now observe that:

- $h$ is non-decreasing in each argument. This is because:

$$\frac{\partial h}{\partial x_i} = h^{1-p} x_i^{p-1} \geq 0.$$

- $h$ is concave. Using the above, we can compute the Hessian $H$ given by:

$$H_{ij} = \frac{\partial^2 h}{\partial x_j \partial x_i} = \begin{cases} (1-p)h^{1-2p}(x_i x_j)^{p-1} & (\text{if } i \neq j) \\ (1-p)h^{1-2p}x_i^{p-2} \cdot (x_i^p - h^p) & (\text{if } i = j) \end{cases}$$

Thus for any $v \in \mathbb{R}^n$, we have:

$$v^T H v = \sum_i \sum_j v_i H_{ij} v_j$$

$$= (1-p)h^{1-2p} \cdot \left( \sum_i v_i \sum_{j \neq i} H_{ij} v_j + \sum_i v_i^2 H_{ii} \right)$$

$$= (1-p)h^{1-2p} \cdot \left( \sum_i v_i x_i^{p-1} \cdot \left( (\sum_j v_j x_j^{p-1}) - v_i x_i^{p-1} \right) + \sum_i v_i^2 (x_i^{2p-2} - h^p x_i^{p-2}) \right)$$

$$= (1-p)h^{1-2p} \cdot \left( (\sum_i v_i x_i^{p-1})^2 - \sum_i (v_i x_i^{p-1})^2 + \sum_i v_i^2 x_i^{2p-2} - \sum_i v_i^2 h^p x_i^{p-2} \right)$$

$$= (1-p)h^{1-2p} \cdot \left( (\sum_i v_i x_i^{p-1})^2 - (\sum_i v_i^2 x_i^{p-2})(\sum_j x_j^p) \right)$$

$$\leq 0,$$

**Algorithm 1** FedBR-BG

---

1: **Input:** Number of iterations in game $H$, number of iterations of gradient descent $T$, learning rate $\alpha$, step size $\delta$, data increasing interval $\Delta s$
2: **Output:** Model weights $\theta^T$, individual contributions $s$
3: **for** $h = 1, 2, \cdots, H$ **do**
4:     Server sends $\theta^t$ to agents;
5:     **for** $t = 0, 1, \cdots, T-1$ **do**
6:         **for** $i \in [n]$ **in parallel do**
7:             $i$ computes $\nabla_{\theta^t} \mathcal{L}_i(\theta^t)$ on its local dataset $\mathcal{D}_i$;
8:             $i$ sends $\nabla_{\theta^t} \mathcal{L}_i(\theta^t)$ to server;
9:         **end for**
10:        Server aggregates the gradients following

$$\nabla_{\theta^t} \mathcal{L}(\theta^t) \leftarrow \frac{1}{\sum_{i \in [n]} |\mathcal{D}_i|} \sum_{i \in [n]} |\mathcal{D}_i| \cdot \nabla_{\theta^t} \mathcal{L}_i(\theta^t);$$

11:        Server updates $\theta^{t+1}$ following

$$\theta^{t+1} \leftarrow \theta^t - \alpha \cdot \nabla_{\theta^t} \mathcal{L}(\theta^t);$$

12:     **end for**
13:     **for** $i \in [n]$ **in parallel do**
14:         $\frac{\partial u_i}{\partial s_i} \leftarrow \frac{a(\sum_i s_i + \Delta s) - a(\sum_i s_i)}{\Delta s} - (1 - \beta) c_i$
15:         **if** $(s_i = 0 \text{ and } \frac{\partial u_i}{\partial s_i} < 0)$ **or** $(s_i = \tau_i \text{ and } \frac{\partial u_i}{\partial s_i} > 0)$ **then**
16:             $s_i^{h+1} \leftarrow s_i^h$;
17:         **else**
18:             $s_i^{h+1} = s_i^h + \delta \cdot \frac{\partial u_i}{\partial s_i}$;
19:         **end if**
20:     **end for**
21: **end for**

---

since $p \leq 1$, $h \geq 0$, and by the Cauchy-Schwarz inequality $(\sum_i a_i \cdot b_i)^2 \leq (\sum_i a_i^2) \cdot (\sum_i b_i^2)$ with $a_i = v_i x_i^{p/2-1}$ and $b_i = x_i^{p/2-1}$. Thus $H$ is negative semi-definite and hence $h$ is concave.

- For each $i$, $g_i(\boldsymbol{b}, \boldsymbol{s}) = u_i(\boldsymbol{b}, \boldsymbol{s})$ is concave.

Using Proposition 2 and the fact that $W_p(\boldsymbol{b}, \boldsymbol{s}) = h(g(\boldsymbol{b}, \boldsymbol{s}))$ we conclude that $W_p(\boldsymbol{b}, \boldsymbol{s})$ is concave.
$\square$

# C   Distributed Algorithms

In this section, we present the distributed algorithms of our two mechanisms, FedBR and FedBR-BG.

**Algorithm 2** FedBR

---

**Input:** Number of iterations in game $H$, number of iterations of gradient descent $T$, learning rate $\alpha$, step size $\delta$, data increasing interval $\Delta s$

**Output:** Model weights $\theta^T$, individual contributions $\boldsymbol{s}$

**for** $h = 1, 2, \cdots, H$ **do**

  Server sends $\theta^t$ to agents;

  **for** $t = 0, 1, \cdots, T - 1$ **do**

    **for** $i \in [n]$ **in parallel do**

      $i$ computes $\nabla_{\theta^t} \mathcal{L}_i(\theta^t)$ on its local dataset $\mathcal{D}_i$;

      $i$ sends $\nabla_{\theta^t} \mathcal{L}_i(\theta^t)$ to server;

    **end for**

    Server aggregates the gradients following

$$\nabla_{\theta^t} \mathcal{L}(\theta^t) \leftarrow \frac{1}{\sum_{i \in [n]} |\mathcal{D}_i|} \sum_{i \in [n]} |\mathcal{D}_i| \cdot \nabla_{\theta^t} \mathcal{L}_i(\theta^t);$$

    Server updates $\theta^{t+1}$ following

$$\theta^{t+1} \leftarrow \theta^t - \alpha \cdot \nabla_{\theta^t} \mathcal{L}(\theta^t);$$

  **end for**

  **for** $i \in [n]$ **in parallel do**

    $\frac{\partial u_i}{\partial s_i} \leftarrow \frac{a(\sum_i s_i + \Delta s) - a(\sum_i s_i)}{\Delta s} - c_i$

    **if** $(s_i = 0$ **and** $\frac{\partial u_i}{\partial s_i} < 0)$ **or** $(s_i = \tau_i$ **and** $\frac{\partial u_i}{\partial s_i} > 0)$ **then**

      $s_i^{h+1} \leftarrow s_i^h$;

    **else**

      $s_i^{h+1} = s_i^h + \delta \cdot \frac{\partial u_i}{\partial s_i}$;

    **end if**

  **end for**

**end for**

---

