# OpenReview forum: "Incentives in Federated Learning: Equilibria, Dynamics, and Mechanisms for Welfare Maximization"
_NeurIPS.cc/2023/Conference — NeurIPS 2023 poster_

### Official Review · Reviewer_KoDx · 2023-07-03

**Soundness:** 3 good
**Presentation:** 3 good
**Contribution:** 4 excellent
**Rating:** 6
**Confidence:** 5

**Summary:**

The paper shows that when every agent has a known concave utility function, the Nash equilibrium exists and can be reached independently via best response updates. Moreover, when all agents have linear costs, the proposed budget mechanism (which modifies the utility function) will lead to Nash equilibrium that maximizes p-mean welfare and fairness across agents.

**Strengths:**

1. The differences with existing works are clearly highlighted. The paper novelly considers (i) the unconstrained setting where agents maximize their concave net utility functions, (ii) optimizing the p-mean welfare of the Nash equilibrium and (iii) the derivation of $\beta^*$.
2. Most of the claims and assumptions of the paper (e.g., concave payoff function, convex cost functions) are well justified.
3. The paper is well-written and easy to follow. The implications described after the theorems aid understanding.

**Weaknesses:**

1. The paper makes two assumptions: a) the cost and payoff functions are common knowledge, and b) any agent utility (payoff and cost) only depends on the number of samples contributed. These assumptions limit the applicability/significance of the work and should be better justified.

For a), Line 317 proposes that the server can train a public accuracy function and broadcast it to all agents. However, in many FL applications, the server may lack data to do so, and clients may be unwilling to share due to the cost incurred. Moreover, the agents may disagree on the relative weightage of the cost/utility function (e.g., how much monetary cost should one unit of payoff function improvement be worth?)

For b), different subsets will lead to models with different performance and costs (e.g., lower if an agent only clones data or one agent may have noisier data). Further justifications (e.g., a trusted data sharing platform ensuring that agent’s i. (or i and j) data is sampled i.i.d) would help.

2. In Sec 4, the definition of $u_i$ is unclear (re-defined on Line 260; 297) and is not explicitly expressed as a function of $p_i$ and $\beta^*$. The descriptions can be improved. My interpretation is that $p_i$ acts as a cost subsidy; hence $u_i = a_i - (c_i - p_i)$ and $(b_i + c_i - p_i = B_i)$. From Theorem 4.1 onwards, $p_i$ uses $\beta^*$ instead of any $\beta$.

**Questions:**

1. Theorem 1 states that the NE using $\beta^*$ will maximize the $p$-mean welfare for any $p ≤ 1$. Does that require the maximizer of the p-mean welfare for every $p ≤ 1$ to be the same? If so, why and how?

One potential counterexample with 2 agents is: out of the utility sample vectors (2,2) and (1,4), (2,2) has higher egalitarian welfare while (1,4) have higher utilitarian welfare.


Minor suggestions (not affecting scores)
* Line 217: explain that $||g(.)||_2 = 0$ at a Nash equilibrium
* Line 255: $c_i$ is used as both the cost function and the per unit cost. The latter should use another notation for less ambiguity.
* Equation after Line 264: add brackets for the summation
* Line 271: spelling ‘defferred’
* Line 289 can be clearer: the maximization problem can be converted to a convex minimization problem.

**Limitations:**

The conclusion briefly identifies the two limitations: 1) the assumption that cost and payoff functions are common knowledge and 2) agents may not report truthfully. The limitations (and weakness 1) can be further elaborated on in the appendix.

---

> ### Author Rebuttal · Authors · 2023-08-10
>
> Thanks for the comments, suggestions and questions.
>
> **Weakness 1:** The paper makes two assumptions: a) the cost and payoff functions are common knowledge, and b) any agent utility (payoff and cost) only depends on the number of samples contributed. These assumptions limit the applicability/significance of the work and should be better justified.
>
> For a), Line 317 proposes that the server can train a public accuracy function and broadcast it to all agents. However, in many FL applications, the server may lack data to do so, and clients may be unwilling to share due to the cost incurred. Moreover, the agents may disagree on the relative weightage of the cost/utility function (e.g., how much monetary cost should one unit of payoff function improvement be worth?)
>
> For b), different subsets will lead to models with different performance and costs (e.g., lower if an agent only clones data or one agent may have noisier data). Further justifications (e.g., a trusted data sharing platform ensuring that agent’s i. (or i and j) data is sampled i.i.d) would help.
>
> **Response:** We will add further justifications on the assumptions. We believe this will help improve the paper, and we thank you for the suggestions.
>
>
> a) We agree that we require costs to be publicly known, or at least verifiable by the mechanism designer. This is a common assumption made in previous work, e.g. Karimireddy et al [1].  Indeed, costs are common knowledge in many real-world applications e.g. (1) in many ML applications, each agent derives their training data from manually labeling a subset of a publicly available dataset like CIFAR or ImageNet, and the cost of labeling dataset is usually known; (2) in autonomous driving, where each data point is a random path taken under random external conditions. Studying incentives when costs are not public is an interesting direction for future work.
>
> Secondly you raise a great point about comparing the scales of cost and utility. However enterprises can (and often need to) correlate the model accuracy and their revenue. For e.g. improved accuracy of a diagnostic test in a hospital via FL will lead to reduced costs in terms of time, salary, and labor of the hospital employees, all of which can be quantified. Likewise for examples (1) and (2) above.
>
> b) This is an excellent point, and studying the model where agent utilities depend on the shared data set (and not just the size of the data set) is an interesting question for future work. Our goal is to make progress on the setting initiated by [1] and [2] where utility functions of the agents depend only on the amount of data shared. This assumption is justified in several practical scenarios, e.g. (1) and (2) above, and we will emphasize this in the paper by incorporating the suggested justification. Thank you so much!
>
> **Weakness 2:** In Sec 4, the definition of $u_i$ is unclear (re-defined on Line 260; 297) and is not explicitly expressed as a function of $p_i$ and $\beta*$. The descriptions can be improved. My interpretation is that $p_i$ acts as a cost subsidy; hence $u_i = a_i - c_i + p_i$ and $b_i + c_i - p_i = B_i$. From Theorem 4.1 onwards, $p_i$ uses $\beta^*$ instead of any $\beta$.
>
> **Response:** Your interpretation of the subsidy $p_i$ is right. We will improve our exposition to make the definitions and intuitive interpretations clearer.
>
> **Question:** Theorem 1 states that the NE using will maximize the $p$-mean welfare for any $p\le 1$. Does that require the maximizer of the p-mean welfare for every $p$ to be the same? If so, why and how? One potential counterexample with 2 agents is: out of the utility sample vectors (2,2) and (1,4), (2,2) has higher egalitarian welfare while (1,4) have higher utilitarian welfare.
>
> **Response:** Our Theorem 1 shows that our mechanism always admits a NE. Secondly, it shows that whenever agents contribute positively at NE $s^*$ (i.e. $s^* > 0$), then $s^*$ also maximizes the $p$-mean welfare among all feasible utility vectors arising from strictly positive data contribution vectors – call the latter set $U^+$. This implies that there is one point in $U^+$ that (weakly) dominates all the other points in $U^+$. Therefore, in your example there should exist a utility vector that dominates both (2,2) and (1,4) and hence that will be the NE as well as p-mean welfare maximizer. We will clarify this in the final version.
>
> References:
>
> [1] Karimireddy et al. “Mechanisms that Incentivize Data Sharing in Federated Learning”, NeurIPS 2022 FL Workshop.
>
> [2] Blum et al. “One for One, or All for All: Equilibria and Optimality of Collaboration in Federated Learning”, ICML 2021.

---

> > ### Comment · Reviewer_KoDx · 2023-08-15
> >
> > Thank you for the clarification and further justification on the assumptions. I acknowledge that I have read the individual rebuttals and your global response.

---

### Official Review · Reviewer_VcZd · 2023-07-03

**Soundness:** 3 good
**Presentation:** 3 good
**Contribution:** 2 fair
**Rating:** 5
**Confidence:** 3

**Summary:**

This paper studies a collaborative Federated Learning framework. Specifically, the authors proposed a utility model, which is the agent’s payoff function minus the data sharing cost function. Under the assumption that the payoff function is concave and the cost function is convex, the authors show the existence of NE. In addition, they proposed a budget-balanced mechanism, which involves payments to the agents, to maximize the agents’ utilities at NE. Experimental results are provided to demonstrate the proposed algorithm's performance, compared with the previous work FedAvg by Blum et al. [2021].

**Strengths:**

Overall, I found the paper to be well-written, and if you take the model as given, the paper gives a fairly satisfying and complete first investigation – the questions asked are exactly the ones I would hope are answered first.

**Weaknesses:**

One major weakness of the paper lies in the utility model proposed. While the paper's key idea of explicitly defining the data-sharing cost is illuminative and contributes to addressing the existence question prevalent in the literature, the utility model itself raises concerns. The critical assumption made in the model is that an agent cannot achieve better utility on its own, even when the cost is zero. Additionally, the assumption suggests that collaboration with the crowd leads to a better utility at the equilibrium state, as opposed to the option of an agent choosing to quit after some rounds of collaboration.

This assumption poses limitations and potentially oversimplifies the complexities of real-world scenarios. It overlooks the possibility that an agent might achieve better utility by pursuing an independent course of action, disregarding the collaborative approach altogether. By not considering the potential benefits an agent could gain from individual efforts, the utility model fails to capture the nuanced dynamics of decision-making and potential trade-offs that exist in real-world situations. Therefore, this utility model undermines the paper's overall conclusions and hinders its practical applicability. I would have liked a more concrete/precise discussion of the motivation for the utility model, for example, what if the cost is too high and the agent prefers to quit the collaboration?

**Questions:**

Can you provide more justification for why the agents always prefer to participate in the collaboration, even if the data-sharing cost is zero when the agent quits the collaboration?

**Limitations:**

N/A.

---

> ### Author Rebuttal · Authors · 2023-08-10
>
> Thank you for the detailed comments.
>
> **Question:** Can you provide more justification for why the agents always prefer to participate in the collaboration, even if the data-sharing cost is zero when the agent quits the collaboration?
>
> **Response:** In a general federated learning framework an agent may not want to participate due to high data sharing cost. However, our mechanism will subsidize such agents (see Line 262), and therefore they may tend to gain by participating compared to the utility they can achieve on their own.
> We also remark that federated learning as a paradigm is most useful for agents who cannot get high utility on their own, and will benefit from the collaboration. For instance, agents having individual data sets on which trained models do not generalize well will benefit from models trained on data from other agents. Naturally, if an agent has extremely high cost of data sharing, for e.g. due to privacy laws, and the payment from our mechanism is insufficient, she will not participate in the collaboration.

---

> > ### Comment · Area_Chair_rmra · 2023-08-18
> >
> > Dear Reviewer vczd,
> >
> > We would appreciate if you could acknowledge and/or respond to the authors' rebuttal.
> >
> > Thank you,
> >
> > AC

---

> > ### Comment · Reviewer_VcZd · 2023-08-18
> >
> > I appreciate the authors' justification, I'm happy to keep my score. I do want to suggest that if the paper is accepted, the author should further consider the practical implications of the results and maybe provide a more thorough discussion in the next version.

---

> > > ### Author Response · Authors · 2023-08-18
> > >
> > > We will definitely add more discussions on the practical implications in the final version. Thank you very much!

---

> > > > ### Author Response · Authors · 2023-08-21
> > > >
> > > > Based on the reviewer's suggestion on having a section on practical implications of our research, we found the following references that seems to provide evidences that agents often benefit through collaboration.
> > > >
> > > > [1] ML models training for threat discovery:  Anna L Buczak and Erhan Guven. 2015. A survey of data mining and machine
> > > > learning methods for cyber security intrusion detection. IEEE Communications
> > > > surveys & tutorials 18, 2 (2015), 1153–1176.
> > > >
> > > > [2] ML models training for drug discovery : Yu-Chen Lo, Stefano E Rensi, Wen Torng, and Russ B Altman. 2018. Machine
> > > > learning in chemoinformatics and drug discovery. Drug discovery today 23, 8
> > > > (2018), 1538–1546.
> > > >
> > > > [3] Data driven  approaches to study the effect of social policies: Victor Chernozhukov, Hiroyuki Kasahara, and Paul Schrimpf. 2021. Causal impact of masks, policies, behavior on early covid-19 pandemic in the US. Journal of
> > > > econometrics 220, 1 (2021), 23–62.
> > > > We will add this discussion to the future revisions of our paper. We are also happy to follow any suggestions that the reviewer may have along this direction

---

### Official Review · Reviewer_NtpC · 2023-07-05

**Soundness:** 4 excellent
**Presentation:** 4 excellent
**Contribution:** 3 good
**Rating:** 5
**Confidence:** 3

**Summary:**

The paper proposes a simple and elegant mechanism for incentivizing data sharing in FL. The mechanism is budget balanced, and any p-mean welfare is maximized at Nash equilibrium. We show the existence of Nash equilibrium (NE) under mild assumptions on agents' payoff and costs. They also show that agents can discover the NE via best response dynamics. In addition, they introduce a FL protocol FedBR-BG based on the budget-balanced mechanism, utilizing best response dynamics. They empirically validate that FedBR-BG outperforms the basic best-response-based protocol without additional incentivization, as well as the standard federated learning protocol FedAvg, in terms of achieving p-mean welfare.

**Strengths:**

The paper introduces public-good-provision approaches to the problem of incentivizing data sharing in FL. They propose a simple and elegant mechanism that achieves desirable properties (based on Falkinger et. al, 2000). The theoretical results are elegant and solid. The presentation is crystal clear.

The results do not seem to rely on any specific feature of FL (not sure whether this is an advantage or disadvantage). There is one result that seems novel and surprising, which is the characterization of the optimal \beta^* (Definition 2) and the fact that it optimizes all p-mean welfare. The proof for the optimality of \beta^* is non-trivial. My evaluation will highly depend on the novelty of this specific result.

**Weaknesses:**

The mechanism seems to be largely based on the previous literature on the public good provision, especially (Falkinger et. al 2000). The mechanism is basically the same as the one in (Falkinger et. al 2000).

The comparison to the existing literature is extremely inadequate. (I didn't see a related work section and the references in the intro barely cover the large literature.) There is a large literature on incentive mechanisms for FL, see [1] for example.

[1] Tu et al., 2021, "Incentive Mechanisms for Federated Learning: From Economic and Game Theoretic Perspective"

**Questions:**

Is the derivation of \beta^* entirely novel, or does it build upon prior work? If it is a novel derivation, why not frame the paper as a study on public good provision? The outcomes seem to have broader implications beyond FL.

Does \beta^* depend on the value of p? Falkinger et. al (2000) seem to suggest that \beta = 1-1/n is optimal for p=1. Does that mean \beta=1-1/n is optimal for all p<= 1?

**Limitations:**

Yes.

---

> ### Author Rebuttal · Authors · 2023-08-10
>
> Thank you for your comments and questions.
>
> **Weakness 1:** The mechanism seems to be largely based on the previous literature on the public good provision, especially (Falkinger et. al 2000). The mechanism is basically the same as the one in (Falkinger et. al 2000).
>
> **Response:** Indeed our mechanism is inspired from Falkinger et al. (2020). The latter assume a simpler model of public goods provision, whereas our model is more general. In Falkinger et al. (2020) (see also [1]), each agent $i$ has a fixed income $B_i$ which is split between private consumption $b_i$ and contribution to the public good $g_i$, and her utility $u_i$ is a function of $b_i$ and $G = \sum_k g_k$, the total monetary contribution to the public good. In our model, agent $i$ spends part of her budget towards the cost $c_i(s_i)$ of sharing $s_i$ data points while $b_i = B_i - c_i(s_i)$ is unspent. Her utility then depends on $b_i$ and $s = \sum_k s_k$, the total data contribution. Our model thus subsumes that of Falkinger et al. when $c_i(s_i) = s_i$ for all $i$.
>
> As far as we know, this intimate connection between FL and public goods provisioning has not been explored before, and is novel. Furthermore, imposing tax/subsidy per agent in our mechanism and that of Falkinger et al. is a common technique used in many other mechanisms for public goods, for eg. see [2]. Our novel contribution is the derivation of the optimal parameter $\beta^*$ for a more general model.
>
> **Weakness 2:** The comparison to the existing literature is extremely inadequate. (I didn't see a related work section and the references in the intro barely cover the large literature.) There is a large literature on incentive mechanisms for FL, see [1] for example. [1] Tu et al., 2021, "Incentive Mechanisms for Federated Learning: From Economic and Game Theoretic Perspective"
>
> **Response:** Thank you for the suggestion. We will add a related work section with detailed discussion of prior works on incentives in FL, including the suggested reference.
>
> **Question 1:** Is the derivation of $\beta^*$ entirely novel, or does it build upon prior work? If it is a novel derivation, why not frame the paper as a study on public good provision? The outcomes seem to have broader implications beyond FL.
>
> **Response:** The derivation of $\beta^*$ is novel, as our model is more general than that of Falkinger et al. We were motivated to address the specific issue of data sharing incentives in FL, but the paper could indeed be more broadly framed as a study on public good provision. We will mention the same in the final version. Again, thank you for the suggestion.
>
> **Question 2:** Does $\beta^*$ depend on the value of $p$? Falkinger et. al (2000) seems to suggest that $\beta = 1-1/n$ is optimal for $p=1$. Does that mean $\beta=1-1/n$ is optimal for all $p \le 1$?
>
> **Response:** We note that $\beta = 1-1/n$ is optimal only when $c_i(s_i) = s_i$ for all $i$ (the result of Falkinger et al.). In general, $\beta^*$ is given by Lemma 6, and depends on the cost functions $c_i$ of the agents, and not on $p$. Our main result has two important elements. First, we show that our mechanism always admits a Nash equilibrium. The second has a technical subtlety: whenever the agents contribute positively at a NE, say $s^*$ (thus $s^* > 0$), then $s^*$ also maximizes the $p$-mean welfare for $p \le 1$ among all utility vectors corresponding to sample vectors where all agents contribute positively. Thus the value $\beta^*$ is optimal for all $p \le1$ whenever the NE has every agent contributing positively. In our specific application, the assumption of “every agent contributing positively” makes sense since an agent can not participate without contributing data in the sense that the server will not communicate with an agent who does not respond!
>
> References:
>
> [1] Falkinger. "Efficient Private Provision of Public Goods by Rewarding Deviations from Average." Journal of Public Economics, 1996.
>
> [2] Andreoni and Bergstrom. “Do Government Subsidies Increase the Private Supply of Public Goods?”, Public Choice, 1996.

---

> > ### Comment · Area_Chair_rmra · 2023-08-18
> >
> > Dear Reviewer ntpc,
> >
> > We would appreciate if you could acknowledge and/or respond to the authors' rebuttal.
> >
> > Thank you,
> >
> > AC

---

> > ### Comment · Reviewer_NtpC · 2023-08-18
> >
> > Thank you for the clarification! I am happy with the response.

---

### Official Review · Reviewer_t7td · 2023-07-12

**Soundness:** 3 good
**Presentation:** 3 good
**Contribution:** 2 fair
**Rating:** 5
**Confidence:** 4

**Summary:**

The authors study the Nash equilibrium in federated learning, under the assumption of concave utility and convex cost, and design welfare maximizing payment for users. Experiments are conducted to verify the mechanism.

**Strengths:**

The topic is of important value, and the paper is well written.

**Weaknesses:**

My major concern is technical novelty and contribution.

1. The analysis of equilibrium and best response dynamic is standard, especially under the assumptions such as convex cost and concave utilities.

2. Though the authors discuss the definition of fairness and welfare maximization, there is lack of discussion on literature about welfare maximization in federated learning, let alone fair subsidies in mechanism design.

3. The design of the payment seems to require the knowledge of the true cost of every user, which is not practical and incentive comptible.

4. The performance of fairness (in Table 1) is compared with methods that are not designed to maximize welfare, which is not meaningful.

**Questions:**

1. More details on the payment (knowledge required, performance comparison with state-of-the-art mechanism) would help.

**Limitations:**

Not applicable.

---

> ### Author Rebuttal · Authors · 2023-08-10
>
> Thank you for your comments and questions.
>
> **Weakness 1:** The analysis of equilibrium and best response dynamic is standard, especially under the assumptions such as convex cost and concave utilities.
>
> **Response:** Using fixed point theorems to prove the existence of Nash equilibrium is a widely used technique in many settings, including under our assumptions of concave payoffs and convex costs. Although these assumptions may seem restrictive, we show that NE need not exist when we go even slightly beyond the concave/convex regime: Theorem A.2 shows an example where no Nash equilibrium exists for agents with non-concave payoffs and linear costs (also stated in see Line 206).
>
> **Weakness 2:** Though the authors discuss the definition of fairness and welfare maximization, there is lack of discussion on literature about welfare maximization in federated learning, let alone fair subsidies in mechanism design.
>
> **Response:** We will add a thorough discussion of prior literature studying welfare maximization in FL. Typically, FL methods compute a model which maximizes some weighted sum of a function of agent’s accuracies; e.g. FedAvg, AFL [1], q-FFL [2]. However these methods ignore data sharing costs and assume agents honestly contribute their available data. Unlike our work, they do not consider the strategic aspects arising from the cost of data sharing.
>
> **Weakness 3:** The design of the payment seems to require the knowledge of the true cost of every user, which is not practical and incentive compatible.
>
> **Response:** We agree that we require costs to be publicly known, or at least verifiable by the mechanism designer. This is a standard assumption made in previous works, e.g. Karimireddy et al [3], and Blum et al. [4] (who assume data sharing cost is known and is linear). It seems, costs are common knowledge in many applications e.g. (1) in many ML applications, each agent derives their training data from manually labeling a subset of a publicly available dataset like CIFAR or ImageNet (2) in autonomous driving, where each data point is a random path taken under random external conditions. Studying incentives when costs are not public is an interesting direction for future work.
>
> **Weakness 4:** The performance of fairness (in Table 1) is compared with methods that are not designed to maximize welfare, which is not meaningful.
>
> **Response:** Table 1 compares the p-mean welfare of our methods, and not fairness. We compare three distributed FL protocols: FedAvg (where agents are not strategic and contribute all data), FedBR (where agents are strategic but there are no payments), and FedBR-BG (our budget balanced mechanism with payments for strategic agents). Both FedBR and FedBR-BG admit Nash equilibria (are fair), while FedAvg ignores strategic aspects of data sharing. We observe from Table 1 that:
>
> (1) FedAvg has lower welfare (i.e. accuracy/payoff minus cost), since agents contribute all data even if it is costly.
>
> (2) When agents are strategic about sharing data but there are no payments, Nash equilibria may not maximize welfare (see the FedBR column of Table 1), and some agents may contribute zero data points (see Example 1 in Sec. 4).
>
> (3) FedBR-BG obtains the highest welfare, thus experimentally supporting our result proving that our mechanism with payments maximizes the welfare of the agents at Nash equilibria.
>
> We also compare with a recent baseline MWFed in [4] on MNIST. MWFed is an adaptation of FedAvg, where the agents adjust their contributions based on the accuracy of the received model. We show that our mechanism outperforms MWFed in terms of p-mean welfare, empirically verifying our theoretical results. Specifically, the p-mean welfare for MWFed is 21.648 (p=0.6), 8.803 (p=0.8), and 5.203 (p=1), in contrast with 24.784 (p=0.6), 9.495 (p=0.8), and 5.340 (p=1) for FedBR, and 25.430 (p=0.6), 9.708 (p=0.8), and 5.459 (p=1) for FedBR-BG.
>
> **Question:** More details on the payment (knowledge required, performance comparison with state-of-the-art mechanism) would help.
>
> **Response:** Thanks for the suggestions for improving this paper. We will include the explanations mentioned above, including (1) details about costs being known, (2) thorough literature survey and comparison to existing methods which maximize welfare but ignore data sharing costs and strategic aspects.
>
> As suggested, we include new experimental results, comparing our method with a recent baseline MWFed of [4], and observe that our mechanism outperforms MWFed in terms of p mean welfare (please refer to the response to weakness 4 above).
>
> We also remark that there is no “state-of-the-art” mechanism for a direct comparison. For instance, Karimireddy et al. [3] compare their mechanism against different self-proposed baseline mechanisms inspired from fairness notions, e.g. proportional data mechanism (which returns to an agent i a model that gives a payoff proportional to number of data points contributed by agent i) (see Appendix C.2 of [3]).
>
> References:
>
> [1] Mohri et al. “Agnostic Federated Learning”, ICML 2019.
>
> [2] Li et al. “Fair Resource Allocation in Federated Learning”, ICLR 2020.
>
> [3] Karimireddy et al. “Mechanisms that Incentivize Data Sharing in Federated Learning”, NeurIPS 2022 FL Workshop.
>
> [4] Blum et al. “One for One, or All for All: Equilibria and Optimality of Collaboration in Federated Learning”, ICML 2021.

---

> > ### Comment · Area_Chair_rmra · 2023-08-18
> >
> > Dear Reviewer t7td,
> >
> > We would appreciate if you could acknowledge and/or respond to the authors' rebuttal.
> >
> > Thank you,
> >
> > AC

---

> > ### Comment · Reviewer_t7td · 2023-08-18
> >
> > I thank the reviewer for the response. I found the response to weakness 3 and the questions is convincing. I am raising my rating to 5. I suggest authors to incorporate these discussions in the revised manuscript.

---

> > > ### Author Response · Authors · 2023-08-18
> > >
> > > Thank you very much! We will incorporate the discussion in the final version.

---

### Author Rebuttal · Authors · 2023-08-10

We thank the reviewers for their time, suggestions and questions that we believe will improve the quality of the paper. Below we summarize our overall response to the reviewer’s questions and comments.

- We discuss our assumption of agent costs being common knowledge. We borrow this assumption from the significant previous works, which seems to be justified in many applications. We described these in detail in our response to Reviewers t7td and KoDx. Thank you for bringing this up, we will add discussion to this end in our final version.

- We will add a discussion comparing other related work studying welfare maximization and incentives in FL, as suggested by Reviewers t7td and NtpC.  In summary, FL methods typically compute a model which maximizes some weighted sum of a function of agent’s accuracies. However these methods ignore data sharing costs and assume agents honestly contribute their available data. Unlike our work, they do not consider the strategic aspects arising from the cost of data sharing.

- We will clarify the novelty about our mechanism and the statement of Theorem 4.1, as suggested by Reviewers t7td, NtpC and KoDx. The derivation of the optimal parameter $\beta^*$ in our mechanism is novel. Moreover, Theorem 4.1 states that the mechanism admits a Nash equilibrium (NE), and whenever agents contribute positively at NE, it also maximizes the p-mean welfare among all utility vectors corresponding to positive sample vectors, for $p\le 1$.

- We perform a new experimental comparison with a relatively recent baseline MWFed from [1]. Our new experiments also show that our mechanism outperforms MWFed in terms of $p$-mean welfare, further supporting our theoretical results. Please refer to the response to Reviewer t7td for the new empirical results.

References:

[1] Blum et al. “One for One, or All for All: Equilibria and Optimality of Collaboration in Federated Learning”, ICML 2021.

---

### Decision · Program_Chairs · 2023-09-21

**Decision:**

Accept (poster)

**Comment:**

This work explores approaches for incentivizing collaborative learning while taking into account both payoffs and costs for each agent. All reviewers agreed that the topic was interesting and timely and that the paper was well-written and executed. It also noted that the analyses approaches themselves are somewhat standard and the assumptions potentially limiting for realistic federated settings. However, as one of the first works rigorously exploring these ideas, the analysis framework and application of it to federated learning seem to be a useful resource for the field, which could potentially be expanded on in future work to consider more complex scenarios. The authors are encouraged to incorporate feedback from the reviewers, especially regarding more thorough discussion of related work (both work on incentives in FL, as well as the discussion of how the proposed mechanism/analysis differ more generally from prior work in mechanism design).